# LEARNING COMPOSABLE CHAINS-OF-THOUGHT

## ABSTRACT

A common approach for teaching large language models (LLMs) to reason is to train on chains-of-thought (CoTs) of in-distribution reasoning problems, but such annotated data is costly to obtain for every problem of interest. We want reasoning models to generalize beyond their training distribution, and ideally to generalize compositionally: they should combine atomic reasoning skills to solve harder unseen tasks. In this paper, we introduce a method to enable generalization to a target compositional task that has no labeled CoT data. We find that simply training models on CoT data of atomic tasks leads to limited generalization, but minimally modifying CoT formats of constituent atomic tasks to be **composable** leads to improvement. Specifically, we augment our data by adding prefixes to CoTs, making sequences of CoTs in-distribution for the trained model. We train individual models on the atomic tasks with composable CoT data and combine them with multitask learning or model merging to address the target compositional task zero-shot. This model can be further trained on a small amount of compositional data using rejection sampling fine-tuning (RFT). Results on three domains of compositional tasks, natural language skills, string manipulation, and arithmetic, show that training LLMs on Composable CoT outperforms multitask learning and continued fine-tuning baselines within a given training data budget.

## 1 INTRODUCTION

Large language models (LLMs) are successful by virtue of the massive amounts of data they are trained on, which makes a wide range of complex problems in-distribution. However, these models still fail at challenging reasoning tasks and it is impossible to scale training data to cover all possible tasks of interest. Ideally, we want models that can *generalize* to new settings, and particularly, can apply basic "skills" learned during training in novel combinations to solve problems at inference time. How to empower LLMs with this capability, also called compositional generalization (Piantadosi & Aslin, 2016; Werchan et al., 2015; Conklin et al., 2021; Dziri et al., 2023), remains an open question. For instance, large reasoning models (QwenTeam, 2025; Guha et al., 2025), built on pre-trained LLMs, are typically trained on a large amount of data annotated with chain-of-thought (CoT) traces, but still fall short at generalizing to harder problem instances than what they were trained on (Sun et al., 2024; Hase et al., 2024; Abreu et al., 2025; Shojaee et al., 2025; Sun et al., 2025).

We explore the setting of compositional reasoning where pre-trained LLMs are fine-tuned on CoT data of simple reasoning tasks (atomic tasks) and then evaluated on the *unseen* combinations of them (compositional tasks) with no or limited compositional supervision. We find that models trained with atomic CoT data demonstrate limited generalization to compositional settings. As illustrated in Figure 1, we propose a simple modification of the CoT format of the atomic task training data, which we call **Composable CoT**: we add "proxy prefixes," which are random filler strings, to the prompt. Then we train models to reason about atomic tasks in the context of the proxy prefixes: this makes the test-time compositional setting more in-distribution, as models need to generate a long reasoning chain by chaining multiple CoTs.

We first experiment with *zero-shot* combination of Composable CoT models. We experiment with two different approaches: first, merging models trained on individual atomic CoT tasks, and second, multitask learning across our atomic CoT datasets. Such combined models achieve zero-shot compositional generalization without seeing compositional data during training.

We then demonstrate that our zero-shot models can be improved further by rejection sampling fine-tuning on a limited amount of compositional supervision. Using *only final answer* supervision, our

Figure 1: A compositional task involves separate atomic skills. We use a data augmentation scheme, **Composable CoT**, to create training data of atomic tasks to teach LLMs CoT formats that can be combined at inference time to address compositional tasks. We augment CoT data to be composable by adding "proxy prefix CoTs," such as random filler strings, to the prompt to simulate compositional distributions where a CoT is conditionally generated from the question and other CoTs.

models can bootstrap better compositional CoT behavior. On tasks involving core reasoning capabilities of LLMs, including string manipulation, arithmetic, and natural language skill composition, our approach outperforms multi-task learning and continued fine-tuning baselines within a given budget of training data. Combining atomic models trained with Composable CoT is consistently better than combining Standard CoT models, with an average performance boost of $18.2\%$ across different compositional tasks.

Moreover, Composable CoT models generalize well to complex compositions with larger skill pools: combining Composable CoT models zero-shot outperforms standard CoT models by an average performance increase of $4.8\%$ on three-way compositions, and $8.8\%$ on two-way compositions that require skill selection.

The main contributions of this work include: (1) A novel data augmentation scheme for training CoT models on basic reasoning skills to enable future composition of them for more complicated reasoning tasks. (2) A method for improving compositional reasoning with LLMs by first training models on atomic CoTs with such augmentation and then performing rejection sampling finetuning.

## 2 PRELIMINARIES

**LLM reasoning with chain-of-thought.** Given a prompt $\mathbf{q}$ that states a reasoning problem with ground truth answer $a$, an LLM $M$ reasons with chain-of-thought by generating a response that includes a chain-of-thought trace $\mathbf{t}$ followed by a predicted answer $\tilde{a}$. Recent works show that supervised fine-tuning pre-trained LLMs on CoT traces leads to strong reasoning models (Muennighoff et al., 2025; Guha et al., 2025). We define a dataset for a task $\mathcal{T}$ with CoT traces of size $N$ as a set of (prompt, CoT, answer) triples: $D_{\mathcal{T}}^{\mathrm{CoT}} = \{(\mathbf{q}, \mathbf{t}, a)\}$.

**Atomic and compositional tasks.** Consider a set of tasks that represent basic reasoning skills, which we call **atomic** tasks. We define **compositional** tasks $\mathcal{T}_A$ as those tasks that can expressed as a composition of $n$ atomic tasks: $\mathcal{T}_A = g(A)$ where $A = \{\mathcal{T}_i, ..., \mathcal{T}_j\}, |A| = n$ and $g$ is some function to combine the $n$ atomic tasks. We discuss more details for $g$ in Appendix A.

**Compositional reasoning from atomic CoT.** We assume access to atomic CoT data $D_A^{\mathrm{CoT}} = \{D_{\mathcal{T}_i}^{\mathrm{CoT}} | \mathcal{T}_i \in A\}$. Models fine-tuned on a subset of $D_A^{\mathrm{CoT}}$ are **atomic CoT models**.

For their composition $\mathcal{T}_A$, we only have access to a training dataset $D_{\mathcal{T}_A}$ of size $N_{\mathcal{T}_A}$. We make two data assumptions following considerations about how compositional data would work in practice: (1) **Answer only:** The data only contains the answers as labels and *not* labeled CoT traces. This reflects that high-quality annotated CoT supervision may be harder to obtain in practice than correct answers; (2) **Limited compositional supervision:** We assume $N_{\mathcal{T}_A}$ is small. We may be able to

collect a small amount of data for each new compositional task of interest, but these compositional tasks are too numerous to undertake large-scale data collection on.

## 3 LEARNING COMPOSABLE CHAINS-OF-THOUGHT

We assume the CoT traces in each of the $n$ atomic task datasets follow a certain distribution distinct to that dataset. A pre-trained LLM $M_0$ fine-tuned on the atomic CoT data can be seen as a mixture model: it can generate CoT traces from each of those $n$ distributions, but it is unclear whether such models can produce compositional CoTs for compositional tasks. We observe that without additional supervision signals, such fine-tuned models typically only replicate one of the learned atomic reasoning patterns in the generated CoT; we show empirical evidence in Section 6.2.

To compose $n$ atomic CoTs in one sequence $\mathbf{t}_1...\mathbf{t}_{n-1}\mathbf{t}_n$, the model must allocate substantial probability to $p(\mathbf{t}_1...\mathbf{t}_{n-1}\mathbf{t}_n \mid \mathbf{q})$, when the model is never trained on a sequence of CoTs. Our goal is to augment the atomic CoT training data $(\mathbf{q}, \mathbf{t}, a)$ into $(\mathbf{q}, \text{proxy prefixes}, \mathbf{t}, a)$, such that **the training data looks more in-distribution to the compositional data while not explicitly training models on compositional examples.**

### 3.1 CONSTRUCTING COMPOSABLE CoT TRAINING DATA

Consider an atomic CoT dataset $D_\mathcal{T}^{\mathrm{CoT}} = \{(\mathbf{q}, \mathbf{t}, a)\}$ for $\mathcal{T} \in A$; we call this **standard CoT** data. We augment it with a set of *chain-of-thought tags* $\mathcal{P} = \{p_k\}$ for $k \in \{1, ..., n\}$.

**Proxy Prefix.** Our goal is to augment standard CoT data such that atomic CoT models can learn a proxy distribution that simulates the distribution of the composition of atomic skills, despite not seeing compositional data. Thus, we append proxy prefixes to the prompt to simulate conditional gener-

| | |
|---|---|
| $\mathbf{q}$ | **Question:** *Multiply the ASCII value of "a" by 2.* |
| $\mathbf{t}_1'$ | *<tag 1> [... aaksebnab  zldjxhl ... ] </tag 1>* |
| ... | *... [Additional proxy prefix CoTs] ...* |
| $\mathbf{t}_{n-1}'$ | *<tag n-1> [dsadu gaulksd ... ] </tag n-1>* |
| $\mathbf{t}_n$ | ***<tag n> The ASCII value of the letter a is 97, and [...] Answer: 194 </tag n>*** |
| $a$ | **Answer: *194*** |

Figure 2: Construction of Composable CoT data. We insert $n-1$ *proxy prefixes*, implemented as sequences of randomly sampled letters, at the end of the prompt, before the CoT.

ation of a CoT given other CoTs. Here we present a simple yet effective approach where the proxy prefix is a sequence of *randomly sampled letters of a random length*. Such a design aims at teaching models to generate robust continuation following an arbitrary prefix CoT. Ablations in Appendix B show that it is more robust to distribution shift than more realistic-looking alternatives.

**Data Construction.** We sample a value of $k$ for each training example $d = (\mathbf{q}, \mathbf{t}, a)$, and we treat $\mathbf{t}$ as the $k$-th step in a notional compositional reasoning process. To achieve this, we append $k-1$ proxy prefixes $(\mathbf{t}_1' \ldots \mathbf{t}_{k-1}')$ to the end of the prompt: $\mathbf{t}_i' =$ *<tag i>*$\mathbf{t}_i$*</tag i>* for $1 \leq i \leq k-1$ and $t_i$ is the $i$-th proxy prefix. By doing so, we obtain the augmented example $d' = (\mathbf{q} \ldots \mathbf{t}_{k-1}', \mathbf{t}_k)$ where $\mathbf{t}_k =$ *<tag k>*$\mathbf{t}$ $a$*</tag k>*.

Figure 2 illustrates the procedure when $k = n$. The standard CoT $\mathbf{t}$ is: "*The ASCII value of the letter a is 97, and [...].*" We augment the example by: (1) Appending $n-1$ proxy prefixes to the end of the question $\mathbf{q}$ to obtain the augmented prompt $\mathbf{q}\mathbf{t}_1' \ldots \mathbf{t}_{n-1}'$, with each proxy prefix wrapped in a tag; (2) Wrapping the CoT and the answer in a different tag *<tag n>* as the augmented response $\mathbf{t}_n$.

We use the scheme above to augment each example in the standard CoT dataset and obtain the augmented dataset $D_\mathcal{T}^{\mathrm{aug}}$. At inference time, we do not know a given atomic CoT will be used in which part of the compositional reasoning trace. Because CoT traces in $D_\mathcal{T}^{\mathrm{aug}}$ can simulate any of the $k$-th positions, models trained on $D_\mathcal{T}^{\mathrm{aug}}$ should be compatible with compositions of *arbitrary order* instead of priming to any particular order seen during training.

**Learning Objective.** Then, we fine-tune $M_0$ on $D_\mathcal{T}^{\mathrm{aug}}$ with a supervised fine-tuning objective: $\mathcal{L}_{D_\mathcal{T}^{\mathrm{aug}}}(\theta) = \frac{1}{N} \sum_{d' \in D^{\mathrm{aug}}} \mathcal{L}_{d'}(\theta)$ where $\mathcal{L}_{d'}(\theta) = -\log p_\theta(\mathbf{t}_k \mid \mathbf{q}...\mathbf{t}_{k-1}')$. In other words, for each augmented example, we minimize the negative log likelihood of generating the CoT and answer, conditioned on the question and the $(k-1)$ proxy prefixes.

---

**Algorithm 1** Bootstrapping Atomic CoT Models Trained on Composable CoT

---

**Input:** The combined model $M_{\text{comb}}$; dataset $D_{\mathcal{T}_A} = \{(\mathbf{q}_v, a_v)\}_{v=1}^{N_A}$; the number of iterations $c$.
**Output:**
1: $M_0 \leftarrow M_{\text{comb}}$                                                       ▷ Initialization
2: **for** $w$ in $1...c$ **do**
3:      **if** use rationalization **then**
4:          $(\tilde{\mathbf{t}}_v, \tilde{a}_v) \leftarrow M_{w-1}(q_v a_v) \,\forall v \in \{1, ..., N_A\}$               ▷ Performance rationalization
5:      **else**
6:          $(\tilde{\mathbf{t}}_v, \tilde{a}_v) \leftarrow M_{w-1}(q_v) \,\forall v \in \{1, ..., N_A\}$
7:      **end if**
8:      $D_{\text{RFT}} \leftarrow \{(\mathbf{q}_v, \tilde{\mathbf{t}}_v, a_v) \text{ s.t. } v \in \{1, ..., N_A\} \text{ and } \tilde{a}_v = a_v\}$       ▷ CoTs with correct answers
9:      $M_w \leftarrow \text{SFT}(M_{\text{comb}}, D_{\text{RFT}})$          ▷ Fine-tune the combined model on the accepted CoT data
10: **end for**

---

Note that when $k = 1$, $d'$ does not have any proxy prefix in the augmented prompt, so the model learns to generate CoT traces conditioned only on the question on those examples (e.g., the top right example in Figure 1). This simulates the scenario where an atomic CoT serves as the initial step of the compositional reasoning. For $1 < k \leq n$, the model learns to generate CoT conditioned on both the question and proxy prefixes (e.g., the bottom right example in Figure 1).

**Instantiation of Tags.** In practice, models only need to learn differentiations between the $n$-th tag, which marks the end of the notional $n$-way compositional reasoning, and all the other tags, which mark intermediate steps. Thus, we set $p_n = \textit{<suffix>}$, and all other $(n - 1)$ tags as $\textit{<prefix>}$. Despite only having two instantiations of the tag, any length of compositional CoT is supported by this scheme.

The scheme can also generalize to $n$-way composition *at inference time*. Specifically, for $n > 2$, we can generate a CoT, then append the $\textit{<suffix>}$ tag, continue to generate, and repeat $(n - 1)$ times, thereby achieving test-time generalization to $n$-way composition. Details can be found in Section 6.1.

## 3.2 COMBINING ATOMIC CoT MODELS

After training an atomic CoT model on a single atomic task $\mathcal{T}$, we need to combine multiple atomic CoT models to perform compositions. We consider two methods.

**ComposableCoT-MTL.** We apply multitask learning (MTL) to fine-tune $M_0$ on the combined dataset of $D_A^{\text{aug}} = \sum_{\mathcal{T}_i \in A} D_{\mathcal{T}_i}^{\text{aug}}$ and obtain a single MTL model $M_{\text{comb}}$ that can generate prefix and suffix CoTs for all the $n$ atomic tasks.

**ComposableCoT-Merge.** Model merging is another way to combine multiple models into a single multi-task model (Matena & Raffel, 2022; Yadav et al., 2023). For each $\mathcal{T}_i \in A$, we start from $M_0$ and fine-tune a model $M_i$ (parametrized by $\theta_i$) on $D_{\mathcal{T}_i}^{\text{aug}}$. Then we use Task Arithmetic (Ilharco et al., 2023a) to merge the $n$ models into a single model $M_{\text{comb}}$ parametrized by $\theta_{\text{comb}}$ as a linear combination of the deltas between each fine-tuned model parameter and the base model parameter: $\theta_{\text{comb}} = \theta_0 + \sum_{\mathcal{T}_i \in A} \alpha_i(\theta_i - \theta_0)$ where $\alpha$ is the scaling factor.

**Inference.** When running zero-shot inferences on the compositional task, we append *<tag 1>* to the end of the prompt and sample a response from $M_{\text{comb}}$. Then, we append the next tag to the end of the generated response, continue generation, and repeat the process by appending tags up to *<tag n>*.

## 3.3 IMPROVING COMPOSITION WITH REJECTION SAMPLING FINE-TUNING

$M_{\text{comb}}$ can be further improved with self-taught reasoning (Zelikman et al., 2022) by rejection sampling fine-tuning (RFT) (Dong et al., 2023; Yuan et al., 2024). Recall that for the compositional task, we only have answer labels instead of CoT traces. $M_{\text{comb}}$ can serve as a starting point for RFT where we fine-tune $M_{\text{comb}}$ with its own correct CoT responses using the limited compositional data.

Algorithm 1 shows the algorithm. Concretely, we sample responses from $M_{\text{comb}}$ for each example in the compositional training data. Using the direct answer labels to verify the sampled responses, we can collect a supervised fine-tuning dataset $D_{\text{RFT}}$ to continued fine-tune $M_{\text{comb}}$. Such a process

Figure 3: Summary of settings for methods evaluated. Names in the results table reference configurations described in this figure; e.g., ComposableCoT-Merge uses ComposableCoTs with model merging, and in the zero-shot setting does not use further tuning.

can be repeated for multiple iterations. For open-ended generation tasks that are hard to verify the correctenss of sampled outputs only based on answer labels, we follow Zelikman et al. (2022); Ye & Durrett (2022) to perform rationalization to obtain $D_{\mathrm{RFT}}$; details can be found in Appendix D.3.

# 4 EXPERIMENTAL SETUP

We select evaluation tasks with the following criteria: (1) **Atomic tasks reflect core LLM reasoning skills**: We select atomic tasks that are representative of core skills that span logical, arithmetic, and writing. Prior work (Wei et al., 2022; Dziri et al., 2023; Yu et al., 2024) has shown that these skills can reflect more complicated capabilities such as advanced math reasoning and creative writing; (2) **Atomic skills are distinguishable**: To ensure controlled experiments of compositional generalization, atomic skills need to be distinguished from each other so that learning one skill is independent from learning another skill; (3) **Compositions are unseen during pretraining**: General reasoning tasks such as math word problems feature examples that are common in pretraining. Our tasks are less observed, thus enabling us to attribute the success of task completion to the efficacy of training approaches rather than better recall of pretraining data.

Our tasks involve string manipulation, arithmetic, and natural language skill composition. Each setting involves atomic tasks and compositional tasks. We ensure that all atomic tasks are learnable through supervised fine-tuning with a small amount of training data ($N_\mathcal{T} \leq 500$) as shown in Appendix E. We also confirm that the selected compositional tasks are less frequently seen for pre-trained LLMs: Appendix F shows the high perplexity of the task datasets, and Table 1 shows the low accuracy of few-shot prompting.

**String manipulation and arithmetic tasks.** We consider the following atomic tasks. **(1) Next letter in alphabet**: Adapted from Efrat et al. (2023); Edman et al. (2024), this task asks the LLM to find the next letter in the alphabet following the last letter in a sequence of letters. **(2) Letter concatenation**: Adapted from Wei et al. (2022); Zhou et al. (2023), this task prompts the LLM to concatenate the first, second, second-to-last, or last letter of each word in a given sequence of words. **(3) ASCII multiplication**: This tasks involves multi-digit multiplicative arithmetic (Dziri et al., 2023; Gambardella et al., 2024) of the ASCII value of a given letter.

We consider the following compositions of two of the atomic tasks, $\mathcal{T}_{(i,j)} = g(\mathcal{T}_i, \mathcal{T}_j)$. We evaluate three-way compositions and more complex compositions in Section 6.1. (1) **Next letter + multiplication**: Given a sequence of letters, find the next letter in the alphabet following the last letter, determine its ASCII value, and then perform multiplication with a given constant. (2) **Concatenation + next letter**: Given a sequence of words, concatenate the first, second, or second-to-last letter of each word and then find the next letter in the alphabet following the last letter of the concatenated sequence. (3) **Concatenation + multiplication**: Given a sequence of words, concatenate the first, second, or second-to-last letter of each word, find the ASCII value of the last letter of the concatenated sequence, and then perform multiplication.

Data and CoT traces of the above tasks are generated with templates; the data generation procedure and examples can be found in Appendix C.

**Natural language skills.** We adapt the compositional benchmark Skill-Mix (Yu et al., 2024): Given the definition and an example of a language skill (e.g. hyperbole), the model needs to write a sentence

Table 1: Zero-shot compositional generalization of ComposableCoT with different combination approaches vs. baselines. *Without any compositional supervision*, using model merging or multitask learning to combine atomic CoT models trained on Composable CoT data outperforms baselines across settings and models, and is sometimes comparable to SFT with compositional supervision.

| Methods | Next Letter + Mult EM | Concat + Next Letter EM | Concat + Mult EM | Skill-Mix Literary + Rhetorical | |
|---|---|---|---|---|---|
| | | | | Full Marks | Skill Fraction |
| Llama 2-7B | | | | | |
| *SFT on Base Model with Compositional Supervision* | 3.1 | 5.0 | 9.0 | 35.5 | 60.1 |
| Few-shot Answer | 1.0 | 0.0 | 0.0 | 4.1 | 16.4 |
| Few-shot CoT | 2.0 | 3.0 | 1.0 | 7.3 | 23.1 |
| StandardCoT-Merge | 2.0 | 12.5 | 2.3 | 11.0 | 31.6 |
| ComposableCoT-Merge (Ours) | 16.0 | **19.1** | 3.0 | 19.6 | 37.1 |
| StandardCoT-MTL | 5.0 | 0.0 | 0.0 | 17.6 | 38.7 |
| ComposableCoT-MTL (Ours) | **18.7** | 6.5 | **3.1** | **22.9** | **49.9** |
| Qwen 2.5-7B | | | | | |
| *SFT on Base Model with Compositional Supervision* | 4.6 | 31.9 | 2.0 | 35.5 | 60.3 |
| Few-shot Answer | 2.4 | 0.0 | 2.7 | 34.7 | 56.0 |
| Few-shot CoT | 2.0 | 0.0 | 21.3 | 31.8 | 41.6 |
| StandardCoT-Merge | 70.4 | 54.8 | **77.0** | 29.8 | 48.0 |
| ComposableCoT-Merge (Ours) | 95.4 | 19.2 | 75.4 | 39.6 | 62.1 |
| StandardCoT-MTL | 3.6 | 60.9 | 72.1 | 42.0 | 58.2 |
| ComposableCoT-MTL (Ours) | **96.3** | **63.3** | 74.3 | **49.0** | **66.7** |

to demonstrate the skill about a given topic. We consider an atomic task to be handling skills over a *category* of skills, and we evaluate on two categories that are mainly mutually exclusive: literary devices (*Literary*) and rhetorical devices (*Rhetorical*). Atomic CoT traces for Skill-Mix are distilled from GPT-4o (OpenAI et al., 2024), following Zhao et al. (2024). The composition tasks we consider combine **literary** and **rhetorical** skills: generate a sentence to demonstrate two provided skills, each of which is sampled from one of the categories. Examples and details can be found in Appendix D.

**Evaluation Metrics.** For Skill-Mix tasks, we use quality measure metrics for the generated sentence from Yu et al. (2024) (namely, *Full Marks* and *Skill Fraction*) based on a rubric, and use GPT-4o-mini as a judge. Details can be found in Appendix D.2. All other tasks are evaluated using *exact match* accuracy; a regex-based answer extractor is used to extract the answer from the generated response.

**Zero-shot/Few-shot Baselines.** Figure 3 summarizes the high-order variables of the configurations we evaluate. For zero-shot compositional generalization, we include the following baselines: (1) Few-shot direct answer prompting: we prompt $M_0$ with 5-shot demonstrations drawn from the compositional data; (2) Few-shot CoT prompting: we prompt $M_0$ with 5-shot CoT demonstrations drawn from the *atomic* data; (3) Model merging of atomic CoT models (*StandardCoT-Merge*): we fine-tune two models $M_i$ and $M_j$ based on $M_0$ with $D_{\mathcal{T}_i}^{\text{CoT}}$ and $D_{\mathcal{T}_j}^{\text{CoT}}$ respectively and merge them into $M_{\text{comb}}$ with Task Arithmetic; (4) Multitask learning of atomic CoTs (*StandardCoT-MTL*): we fine-tune $M_0$ to be a single multitask learning model $M_{\text{SCoT}-\text{MTL}}$ on $D_{\mathcal{T}_i}^{\text{CoT}} + D_{\mathcal{T}_j}^{\text{CoT}}$.

**Baselines with Compositional Supervision.** With the *same* compositional training dataset with only the answer label $D_{\mathcal{T}_{(i,j)}}$, we compare bootstrapping Composable CoT with the following baselines. (1) Continued fine-tuning (CFT) the multitask model of atomic CoTs (*CFT on StandardCoT-MTL*): we continue fine-tune the multitask model $M_{\text{SCoT}-\text{MTL}}$ on $D_{\mathcal{T}_{(i,j)}}$; (2) Continued fine-tuning the merged model of atomic CoTs (*CFT on StandardCoT-Merge*): we continue fine-tune the merged model of the two atomic CoT models $M_{\text{comb}}$ on $D_{\mathcal{T}_{(i,j)}}$; (3) Multitask learning of atomic CoTs and compositional answers (*StandardCoT + Comp Answer*): we fine-tune a single multitask learning

Table 2: Compositional task performance of rejection sampling fine-tuning (RFT) upon merged Composable atomic CoT models and other baselines. *Mult* stands for ASCII multiplication and *concat* stands for letter concatenation. *SFT* stands for supervised fine-tuning with the compositional answer data; *CFT* stands for continued fine-tuning; *MTL* stands for multitask learning method. Results on next letter + mult are omitted because the zero-shot performance saturates. RFT on ComposableCoT variants achieves the best compositional performance using the same compositional answer data.

| Category | Method | Next Letter + Mult EM | Concat + Next Letter EM | Concat + Mult EM | Skill-Mix Literary + Rhetorical Full Marks | Skill Fraction |
|---|---|---|---|---|---|---|
| | | | Llama 2-7B | | | |
| SFT | SFT on Base Model | 3.1 | 5.0 | 9.0 | 35.5 | 60.1 |
| | CFT on StandardCoT-Merge | 2.0 | 16.0 | 14.0 | 44.1 | 65.1 |
| | CFT on StandardCoT-MTL | 3.0 | 26.0 | 11.0 | 38.0 | 62.1 |
| MTL | StandardCoT + Comp Answer | 5.0 | **46.0** | 13.3 | 22.9 | 45.5 |
| RFT | StandardCoT-Merge | 0.0 | 23.0 | 29.7 | 26.1 | 52.0 |
| | ComposableCoT-Merge (Ours) | **72.0** | **46.0** | **40.0** | **45.3** | **66.6** |
| | | | Qwen 2.5-7B | | | |
| SFT | SFT on Base Model | - | 31.9 | 2.0 | 35.5 | 60.3 |
| | CFT on StandardCoT-Merge | - | 41.1 | 9.3 | 51.0 | 71.4 |
| | CFT on StandardCoT-MTL | - | 60.3 | 12.7 | 34.7 | 56.3 |
| MTL | StandardCoT + Comp Answer | - | 65.1 | 7.1 | 41.2 | 55.3 |
| RFT | StandardCoT-MTL | - | 82.1 | **89.0** | 44.9 | 63.4 |
| | ComposableCoT-MTL (Ours) | - | **86.9** | 88.4 | **57.6** | **71.5** |

model based on $M_0$ on the combined dataset of $D_{\mathcal{T}_i}^{\mathrm{CoT}} + D_{\mathcal{T}_j}^{\mathrm{CoT}} + D_{\mathcal{T}_{(i,j)}}$. We also include supervised learning baselines (SFT) where $M_0$ is fine-tuned on the same compositional answer data $D_{\mathcal{T}_{(i,j)}}$.

The differences of methods we evaluate for each setting are summarized in Table 13.

**Data Construction.** Because of two-way compositions, we sample uniformly from 2 chain-of-thought tags, *<prefix>* and *<suffix>*, for data construction. At inference time, we first append *<prefix>* to the prompt and sample from the combined model. Then, we append *<suffix>* to the end of the generated response, and continue generation.

**Models and Training.** We use Llama 2 7B-base (Touvron et al., 2023) and Qwen2.5 7B-base (Yang et al., 2025) as models, and use LoRA (Hu et al., 2022) for supervised fine-tuning. For rejection sampling, we sample 10 responses for each prompt and use temperature $\tau = 0.9$ for inference; otherwise, we use greedy decoding. For Skill-Mix tasks, we perform rationalization for RFT because it is open-ended generation (see Section 3.3). Configuration and hyperparameters are in Appendix G.

## 5 RESULTS

### 5.1 ZERO-SHOT GENERALIZATION

We evaluate the compositional generalization of the proposed method *without compositional supervision*, including ComposableCoT-Merge and ComposableCoT-MTL. For all methods that we compare with, we control the amount of training data to be the same as $N_i$ and $N_j$. For reference, we also include the supervised fine-tuning baseline by fine-tuning $M_0$ with $N_{(i,j)}$ compositional answer data. Details of the training data for each task can be found in Appendix H.

**Learning ComposableCoT achieves better zero-shot generalization.** Table 1 shows that ComposableCoT variants outperform all baselines on a range of tasks for both models. Combining atomic CoT models trained on Composable CoT is better than combining models trained on standard CoT across settings. Moreover, while having seen no compositional training data, our method achieves comparable or even better performance than supervised fine-tuning baselines *with* compositional

supervision (e.g., next letter + multiplication). These indicate that the Composable CoT format leads to better "composability" at inference time.

## 5.2 Compositional Performance with Limited Supervision

We evaluate the performance of Composable CoT models after being further improved with one iteration of RFT using the limited compositional supervision. We compare it with multitask learning and continued fine-tuning baselines given the same compositional answer dataset $D_{\mathcal{T}_{(i,j)}}$ of size $N_{(i,j)} \leq 500$. For reference, we include the baseline of fine-tuning $M_0$ on the same compositional answer data. Details of the data condition are in Appendix H.

Table 3: Zero-shot generalization on three-way compositions. Combining ComposableCoT models outperforms combining StandardCoT models on the composition of three tasks.

|  | String Tasks | Skill-Mix | |
| --- | --- | --- | --- |
|  | EM | Full Mark | Skill Fraction |
| Standard-Merge | 61.3 | 13.1 | 42.7 |
| Composable-Merge | 63.1 | 19.2 | 54.1 |
| Standard-MTL | 82.3 | 28.2 | 55.9 |
| Composable-MTL | **86.7** | **33.1** | **61.0** |

Table 2 shows that with the same compositional training data, **using RFT on top of ComposableCoT-MTL and ComposableCoT-Merge achieves the best compositional task performance**, outperforming multitask learning and continued fine-tuning baselines across settings.

We further investigate if the performance is mainly driven by RFT or by learning Composable CoT format. We compare RFT upon StandardCoT-Merge with RFT upon ComposableCoT-Merge for LLama 2-7B, and StandardCoT-MTL with ComposableCoT-MTL for Qwen 2.5-7B. Table 2 shows that RFT is a better way to improve the compositional task performance of StandardCoT models with compositional data than MTL and SFT. Moreover, **RFT upon Com-**

Table 4: Zero-shot generalization on two-way compositions when merging **three** atomic models (i.e., there is a distractor skill). Merging ComposableCoT models is better than merging StandardCoT models in this setting.

|  |  | Standard | Composable |
| --- | --- | --- | --- |
| Next Letter + Mult | EM | 56.1 | **75.9** |
| Concat + Next Letter | EM | 39.1 | **46.2** |
| Concat + Mult | EM | 44.3 | **48.9** |
| Skill-Mix Literary | Full Mark | 37.1 | **42.0** |
| + Rhetorical | Skill Fraction | 55.1 | **62.7** |

**posableCoT models is generally better than RFT upon StandardCoT models.** Using the same combination method (MTL or Model Merging), RFT upon ComposableCoT models outperforms the StandardCoT counterpart by an average performance increase of $18.2\%$ across models and tasks.

## 6 Analysis

### 6.1 Generalization to Complex Compositions

**Three-way Composition.** We evaluate Composable CoT on zero-shot compositions of *three* atomic tasks on Qwen2.5-7B using the following compositional tasks: (1) Letter Concat + Next Letter + Mult (*String Tasks*): Given a sequence of words, concatenate the first, second, second-to-last, or last letter of each word, find the next letter in the alphabet following the last letter of the concatenated sequence, find the ASCII value of this letter, and then perform multiplication. (2) Skill-Mix Literary + Rhetorical + Logical (*Skill-Mix*): Generate a sentence on a given topic to demonstrate three provided skills, each of which is sampled from one of the Skill-Mix categories, including an additional category *Skill-Mix-Logical*. We compare ComposableCoT models with StandardCoT models constructed by model merging or multi-task learning. Implementation details can be found in Appendix I.

Table 3 shows that given the same combination method, combining ComposableCoT models is better on three-way composition: for example, using MTL, ComposableCoT models outperform StandardCoT models by an average performance increase of $4.8\%$.

**Two-way Composition with Larger Skill Pools.** In practice, models may need to have many capabilities to address problems of interest. Compared to our existing settings, such models need to select the skills to engage with for a particular task out of a larger pool of learned skills.

To evaluate this scenario, we train atomic models on *three* atomic skills on Qwen 2.5-7B and evaluate the combined model (with model merging) on the zero-shot composition of *two* of the atomic skills. This setup provides the model with additional potential combinations of skills to reason about. For the Skill-Mix tasks, we train on an additional atomic task, logical reasoning. Table 4 shows that learning ComposableCoT outperforms using StandardCoT by $8.8\%$ on average, indicating that ComposableCoT models can select the appropriate skills to compose out of many learned skills.

## 6.2 QUALITY OF GENERATED COTS

We conduct intrinsic quality evaluations on CoTs generated by ComposableCoT models for zero-shot composition. For the string manipulation and arithmetic tasks, we extract template-based patterns of each atomic CoT from the generated outputs of models evaluated on the compositional task. For Skill-Mix, we consider the CoT pattern of an atomic task to be used if the generated response explicitly mentions the skill corresponding to that atomic skill category.

Table 5 shows results with models trained from Qwen 2.5-7B and combined with MTL; results using model merging can be found in Appendix J. **Combining ComposableCoT leads to consistently higher presence of both atomic CoT patterns in the generated responses compared to StandardCoT.** Models trained with the Composable CoT format therefore leverage the combination of learned skills in some form more frequently than StandardCoT. Example of generated CoTs can be found in Appendix K.

Table 5: Quality of the generated CoTs by ComposableCoT models on zero-shot compositions. "% $\mathcal{T}_1$" denotes the percentage of generated responses that use the CoT format of the first atomic task of the composition, and likewise for the second. [†] denotes that the ComposableCoT method has a significantly higher "% Both" than the StandardCoT counterpart at the $0.01$ level using a paired bootstrap test. "Perf." denotes the task performance.

| | CoT | Perf. | % $\mathcal{T}_1$ | % $\mathcal{T}_2$ | % Both |
|---|---|---|---|---|---|
| Next Letter | Standard | 3.6 | 0.0 | 100.0 | 0.0 |
| + Mult | Composable | 96.3 | 98.9 | 100.0 | [†]98.9 |
| Concat | Standard | 72.1 | 99.7 | 32.1 | 32.1 |
| + Next Letter | Composable | 74.3 | 100.0 | 83.1 | [†]81.3 |
| Concat | Standard | 60.9 | 100.0 | 66.7 | 66.7 |
| + Mult | Composable | 63.3 | 100.0 | 85.9 | [†]85.0 |
| Literary | Standard | 42.0 | 65.3 | 58.0 | 37.6 |
| + Rhetorical | Composable | 49.0 | 64.5 | 65.7 | [†]42.0 |

## 7 RELATED WORK

As an important cognitive capability of humans (Piantadosi & Aslin, 2016; Werchan et al., 2015), compositional generalization has been considered a core capability for human-level reasoning models (Fodor & Pylyshyn, 1988; Lake & Baroni, 2023). Recent theoretical analyses show that LLMs can improve their compositional reasoning by generating CoT (Li et al., 2024; 2023), but empirical improvements have only been observed (Sprague et al., 2025) with non-trivial engineering effort such as prompt engineering (Chen et al., 2024; Gao et al., 2024) and data selection (Khot et al., 2023; Zhou et al., 2023; Levy et al., 2023; Ye et al., 2023). Aiming at more principled ways to improve composition, we are inspired by a line of work on efficient methods for combining models of different capabilities, particularly model merging (Tam et al., 2024a; Ilharco et al., 2023b; Wu et al., 2025; KimiTeam et al., 2025; Ma et al., 2025; Tam et al., 2024b). Our work is the first to use model merging for compositional generalization with CoT.

## 8 CONCLUSION

We propose Composable Chain-of-Thought, a data augmentation scheme to convert CoT data of atomic reasoning skills into a format that facilitates inference-time compositional generalization. Training atomic CoT models with Composable CoT and combining them with model merging or multitask learning leads to better zero-shot compositional reasoning performance than building models with the standard CoT format. Such a combined model can be further improved by a limited amount of compositional data with rejection sampling fine-tuning. Learning to reason with composable CoT shows a promising approach to improve compositional reasoning in LLMs, and could be extended to build more efficient and robust large reasoning models.

## 9 ETHICS STATEMENT

This work does not involve human subjects or the release of sensitive data. We do not clearly see the harms of the applications of the proposed method either, so we are not aware of any obvious ethical concern related to this work.

## 10 REPRODUCIBILITY STATEMENT

We report all technical details for our proposed method, including the data augmentation schema and the training methods in Section 3. To reproduce our experimental results, we report all details of the evaluation setup (Section 4) and training configurations (Section G).

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

## A  A NOTE ON COMPOSING TASKS

There exist various possible ways to combine atomic tasks into a compositional task with the combination function $g$. We simplify $g$ into two types: (1) composite: the output of one atomic task is used as part of the input of another task, $g(\mathcal{T}_i, \mathcal{T}_j) = \mathcal{T}_i \circ \mathcal{T}_j$ or $g(\mathcal{T}_i, \mathcal{T}_j) = \mathcal{T}_j \circ \mathcal{T}_i$; (2) concatenation: the outputs of the two atomic tasks are concatenated using the same input, $g(\mathcal{T}_i, \mathcal{T}_j) = \mathcal{T}_i \oplus \mathcal{T}_j$ or $g(\mathcal{T}_i, \mathcal{T}_j) = \mathcal{T}_j \oplus \mathcal{T}_i$. Among tasks evaluated in Section 4, the string manipulation and arithmetic tasks need to be solved by a composite function, while the Skill-Mix task can be solved by either a composite function or a concatenation function.

## B  DESIGN CHOICES FOR CONSTRUCTING COMPOSABLE CoT DATA

When designing the proxy prefix CoT, we would like to consider the following constraints. (1) We do not assume any prior knowledge about what would possibly be put in the proxy prefix at inference time; (2) We do not assume strong relevance between the proxy prefix CoT and the actual CoT, i.e., not all the information in the proxy prefix CoT is useful for predicting the CoT and the final answer. Based on these considerations, we experiment with the following variants:

- **Random letters**: We sample random letters from the alphabet to form a sequence of random lengths to simulate an *arbitrary* prefix CoT.
- **Random text from the prompt**: We sample random letters and words from the prompt $\mathbf{q}$ to form a sequence of random lengths to simulate a prefix CoT in a similar distribution as the input distribution.

Table 6: Performance of atomic CoT models fine-tuned on different variants of proxy prefix on Llama 2-7B. Using random letters as the proxy prefix achieves the best out-of-domain performance when evaluated with an unseen prefix at inference time.

| Type of Proxy Prefix | Exact Match Accuracy | |
| --- | --- | --- |
| | In Domain Prefix | Out-of-Domain Prefix |
| Random Letters | 83.0 | 90.0 |
| Random Text from the Prompt | 86.4 | 82.5 |
| Random Text from Web | 90.6 | 70.0 |

- **Random text from web**: We sample random sentences from OpenWebText (Gokaslan & Cohen, 2019) to simulate a prefix CoT drawn from the pretraining data distribution.

We evaluate these variants by fine-tuning models on Composable CoT datasets **that only the following augmentation**: $d' = \mathbf{q}$ *<prefix>* [proxy prefixes] *</prefix> <suffix>* $\mathbf{t}a$ *</suffix>*. Note that this is different from the setting discussed in Section 3.1 where the Composable CoT dataset consists of other possible augmentations as well based on the sampling of the tags (e.g., $d' = \mathbf{q}$ *<prefix>* $\mathbf{t}a$ *</prefix>* when $k = 1$). This experiment mainly aims at stress testing the model's capability of learning a single atomic task with a given proxy prefix CoT variant. We use the same hyperparameter configurations for all proxy prefix variants for a given task.

We evaluate the fine-tuned models on the in-domain task in two settings: (1) *In-domain prefix*: we append the same type of prefix as we have used for training to the end of the prompt of the in-domain test example and evaluate the model on it; (2) *Out-of-domain prefix*: we randomly sample a prefix from the other two variants and append it to the end of the prompt of the in-domain test example and evaluate the model on it. We run experiments on the three string manipulation and arithmetic tasks and report the average performance. Table 6 shows that although using random letters as the proxy prefix leads to the worst in-domain performance, it generalizes the best to out-of-domain prefixes, which is a more desirable behavior.

## C  DETAILS OF STRING MANIPULATION AND ARITHMETIC TASKS

**Next letter in alphabet**   We synthetically generate data for Next letter in alphabet. We randomly sample letters from the English alphabet of a random length and concatenate them into a sequence. Then we extract the last letter from the sequence and derive the next letter following it in the alphabet. An example can be found in Example C.2. We automatically generate a chain-of-thought for each generated problem, using a fixed template shown in Example C.2.

**ASCII multiplication**   Similarly, we randomly sample letters from the English alphabet of a random length and concatenate them into a sequence. Then, we randomly sample another letter $s$ and randomly sample an integer $a \in \{1, ..., 9\}$. We find the ASCII value of $s$ as $f(s)$ and compute the product $af(s)$ as the gold answer. An example can be found in Example C.3. We automatically generate a chain-of-thought for each generated problem, using a fixed template shown in Example C.3.

**Letter concatenation**   We follow Wei et al. (2022) to generate the dataset by randomly sampling from the most popular first and last names in the United States and the United Kingdom from `https://namecensus.com` and randomly concatenating them into a sequence of names. While the original task in Wei et al. (2022) only requires concatenating the last letter of each name together, we raise the difficulty level by randomly asking for concatenations of the first, second, second-to-last, or the last letter. An example can be found in Example C.1. The CoT template is also shown in Example C.1.

**Compositional tasks**   We synthetically construct the compositional tasks of the string manipulation and arithmetic tasks in similar procedures as used to generate the atomic data. An example of next letter + ASCII multiplication can be found in Example C.4, concatenation + next letter in Example C.5, and concatenation + multiplication in Example C.6. We made a design decision to

exclude one variant of concatenation + next letter that concatenates the last letter of each word and finds the next letter following the last letter in the concatenated sequence; this variant can be solved by the reasoning shortcut of only applying Next letter in alphabet rather than a composition of both.

---

### C.1 Atomic Task Example: Letter Concatenation Example

```
[Instruction]
Take the second-to-the-last letter of each word in the sequence
and concatenate them in lower case:  Tequan Monjur Khia
Jodi-leigh answer

[Chain-of-Thought + Answer String]
The second-to-the-last letter of the 1st word is a.  The
second-to-the-last letter of the 2nd word is u.  The
second-to-the-last letter of the 3rd word is i.  The
second-to-the-last letter of the 4th word is g.  So the answer
is auig.

[Answer String]
auig
```

---

### C.2 Atomic Task Example: Next letter in alphabet

```
[Instruction]
Find the Next letter in alphabet following the last letter in
the sequence:  wqsisibnnicdlpwqbnoicdcxcxrfoilpcbnixuc
bssssejxuzods answer:

[Chain-of-Thought + Answer String]
The last letter is s, and the letter following it in alphabet is
t.  So the answer is t.

[Answer String]
t
```

---

### C.3 Atomic Task Example: ASCII Multiplication

```
[Instruction]
Find the ASCII value of the letter after '<letter>' and multiply
the ASCII value by 2:  byaxaxcpoteznwnwseselyjlretx
txcbfvmfezbycplymfotjbfv
jlhotzjbjcpycbtzhorepyjckofj <letter> d answer:

[Chain-of-Thought + Answer String]
The ASCII value of the letter d is 100, and multiplying the
ASCII value by 2 gives us 200.  So the answer is 200.

[Answer String]
200
```

---

### C.4 Compositional Task Example: Next letter + ASCII Multiplication

```
[Instruction]
Find the ASCII value of the Next letter in alphabet following
the last letter in the sequence and multiply the ASCII value by
5:  knnxqsxvshqugxfuquljumsbihgxvqihnxuufuknxvumuupkpkshljqsbkiz
answer:
```

---

```
[Answer String]
485
```

## C.5 Compositional Task Example: Concatenation + Next Letter

```
[Instruction]
Take the second-to-the-last letter of each word in the sequence,
concatenate them in lower case, and find the Next letter in
alphabet following the last letter in the sequence of the
concatenated letters:  Tyjai Ahijah Denzil Amine answer:

[Answer String]
o
```

## C.6 Compositional Task Example: Concatenation + Multiplication

```
[Instruction]
Take the second-to-the-last letter of each word in the sequence,
concatenate them in lower case, then find the ASCII value of
the last letter in the sequence of the concatenated letters,
and multiply the ASCII value by 3:  Zarriah Amylee Li Javarie
answer:

[Answer String]
315
```

# D  DETAILS OF SKILL-MIX TASKS

## D.1  MODIFICATIONS OF SKILL-MIX

We adapt the Skill-Mix dataset from Yu et al. (2024). For each example, the model is given a natural language skill, its definition, an example of the skill, and a topic to focus on, and the model needs to write a grammatical sentence to demonstrate the skill on the topic. Because we mainly focus on pairwise composition, we only use the $k = 2$ and $k = 1$ composition sets of the Skill-Mix data. We apply the following modifications to the dataset to fit our setting of compositional reasoning.

1. Filtering the categories of skills: We keep examples with skills of the rhetorical and literary categories out of the five categories from the original dataset. This is because the rhetorical and literary skills have the least overlap while other categories have more (e.g. the logical and rhetorical skills have a large body of overlaps).

2. Removing the requirements of post-hoc explanation and refinement from the prompt. The original dataset evaluates models by prompting the models to first write a sentence, provide an explanation for the written sentence, and then do another round of refinement based on feedback from the grader (an LLM-as-a-judge). To fit into our setting of chain-of-thought reasoning and direct answering, we remove these irrelevant elements in the prompt.

3. Using a public test set: The original evaluation of Yu et al. (2024) hides the test set from the public and models can only be evaluated based on API calls to the hidden test set. To ensure reproducibility of our results, we use a public test set collected by Zhao et al. (2024).

As an open-ended generation task, Skill-Mix does not have a single ground truth sentence. Zhao et al. (2024) collect a silver-labeled dataset of sentences and their post-hoc explanations generated by GPT-4o (OpenAI et al., 2024) and graded as correct by the grader. We further prompt GPT-4o to transform the (sentence, explanation) pairs into (CoT, sentence) pairs to resolve potential coreference issues. Examples of the prompt, CoT traces, and sentence can be found in Examples D.1, D.2, and D.3.

### D.1 Atomic Task Example: Skill-Mix-Literary

```
[Instruction]
Greetings!  I am interested in natural language processing and
I was wondering if you could help me generate an example of text
that illustrates multiple skills in semantics or syntax.  The
example should be a minimal natural piece of text with up to a
few lines in the context of Vikings that illustrates all of the
following skills:  oxymoron.  Please keep the text as short as
possible, and make sure the concepts can be found fully from the
text.

For reference, here are the definitions and examples for the
concepts:
**oxymoron**:  A figure of speech in which apparently
contradictory terms appear in conjunction.  For example, "The
hall was filled with a deafening silence."  Here, the phrase
"deafening silence" is an oxymoron.

Please start the minimal natural piece of text with 'Answer:'.

Thanks very much!

[Chain-of-Thought + Answer String]
Explanation:  The goal is to construct a sentence that showcases
the use of an oxymoron while remaining relevant to the Viking
theme.  To achieve this, we focus on creating a phrase that
contains seemingly contradictory terms.  We begin with the
context of Vikings enduring harsh weather conditions.  The
word "fiery" conveys an intense heat, while "frost" indicates
severe cold.  When these terms are combined into "fiery frost,"
they form an oxymoron as they depict both extreme temperature
sensations simultaneously.  This juxtaposition effectively
creates a semantic paradox, leading to vivid imagery and
emphasizing the harsh conditions faced by Vikings.

Answer:  "In the bitter cold, the Viking felt a fiery frost."

[Answer String]
Answer:  "In the bitter cold, the Viking felt a fiery frost."
```

### D.2 Atomic Task Example: Skill-Mix-Rhetorical

```
[Instruction]
Greetings!  I am interested in natural language processing and
I was wondering if you could help me generate an example of
text that illustrates multiple skills in semantics or syntax.
The example should be a minimal natural piece of text with up
to a few lines in the context of Hiking that illustrates all
of the following skills:  begging the question or assuming the
conclusion.  Please keep the text as short as possible, and make
sure the concepts can be found fully from the text.

For reference, here are the definitions and examples for the
concepts:
**begging the question or assuming the conclusion**:  A circular
argument that paraphrases the question.  For example, "Of course
the earth is round; people have known this for hundreds of
```

```
years."

Please start the minimal natural piece of text with 'Answer:'.

Thanks very much!

[Chain-of-Thought + Answer String]
Explanation:  To illustrate the concept of begging the
question or assuming the conclusion within the context of
hiking, we start by identifying a statement that relies on
its own conclusion as a premise.  First, recognize that the
claim "hiking is beneficial" needs supporting evidence to be
considered valid.  However, if we use the phrase "because it's
good for your health" as justification, we notice that it merely
reiterates the initial claim using slightly different wording.
This is because declaring something "beneficial" inherently
implies a positive impact, such as being "good for your health."
Thus, the reasoning becomes circular, as it depends on the same
assumption it seeks to prove.

Answer:  "Hiking is beneficial because it's good for your
health."

[Answer String]
Answer:  "Hiking is beneficial because it's good for your
health."
```

---

**D.3 Compositional Task Example: Skill-Mix Literary + Rhetorical**

```
[Instruction]
Greetings!  I am interested in natural language processing and
I was wondering if you could help me generate an example of text
that illustrates multiple skills in semantics or syntax.  The
example should be a minimal natural piece of text with up to a
few lines in the context of Vikings that illustrates all of the
following skills:  anaphora resolution, begging the question
or assuming the conclusion.  Please keep the text as short as
possible, and make sure the concepts can be found fully from the
text.

For reference, here are the definitions and examples for the
concepts:
**anaphora resolution**:  Resolving the antecedent of a pronoun
or noun phrase.  For example, "The car is falling apart, but
it still works."  Here , "it" is the anaphor and "car" is the
antecedent.
***begging the question or assuming the conclusion**:  A
circular argument that paraphrases the question.  For example,
"Of course the earth is round; people have known this for
hundreds of years."

Please start the minimal natural piece of text with 'Answer:'.

Thanks very much!

[Answer String]
Answer:
```

```
The Viking chief, undefeated thanks to his ship, asserted, "It
remains unconquered because it's the 'Indomitable'."
```

## D.2 EVALUATION METRICS

We use GPT-4o-mini as the LLM-as-a-judge to grade the generated sentence using the exact same grading rubric as provided by Yu et al. (2024); the grader judges the quality of the sentence based on if: (1) All skills are used; (2) The sentence makes sense; (3) The sentence attaches to the given topic; (4) The sentence is short. We use the evaluation metrics for each generated sentence in Yu et al. (2024), including the following:

1. **Full Marks:** 1 if the generated sentence satisfies all four criteria above and 0 otherwise.

2. **Skill Fraction:** The fraction of skills being demonstrated if all the other three criteria are satisfied; 0 otherwise

We aggregate these metrics by averaging over all generated responses. In general, full marks evaluate the model's capability of writing a perfect sentence for the task, while skill fraction evaluates how good the model is at handling skills given that it is good at the other stylistic capabilities. We use Curator (Marten et al., 2025) for an efficient implementation of the evaluation pipeline.

## D.3 RFT ON SKILL-MIX TASKS WITH RATIONALIZATION

For open-ended generation tasks like Skill-Mix, it is hard to only use the reference answer to verify the correctness of the responses sampled from a model. Thus, we use rationalization to perform rejection sampling fine-tuning for Skill-Mix: we first append the direct answer label to the end of the prompt and sample post-hoc explanations for the given answer from the model; because $M_{\text{comb}}$ is optimized to generate an answer following a CoT, we extract the generated answer following the generated explanation and filter out explanations whose following answer is not the same as the provided gold answer; finally, we use the accepted explanations as surrogates for CoT to form the RFT data.

## E  SINGLE-TASK LEARNING PERFORMANCE

We report the single-task learning performance of the atomic CoT models by evaluating them on the in-domain atomic tasks. We would like the atomic tasks to be easy to learn to reflect the practical settings where we train models on basic, easy-to-learn skills and generalize to harder, unseen tasks. The training data conditions and hyperparameters for training can be found in Appendix G. Table 7 shows that all atomic tasks we evaluate are learnable within a small amount of training data $(N_i, N_j \leq 500)$.

In addition, we observe that training on ComposableCoT or StandardCoT does not lead to consistent differences in atomic CoT performance, while the exception is on Skill-Mix-Rhetorical for Llama 2-7B where fine-tuning on ComposableCoT outperforms fine-tuning on StandardCoT by a large margin.

## F  BASE MODEL PERFORMANCE ON EVALUATION TASKS

To confirm that our task design includes evaluation tasks that are less commonly seen in the pretraining data of LLMs, we evaluate the perplexity of the task datasets.

We compare the datasets of the string manipulation and arithmetic datasets used in our experiments with mathematical reasoning data (GSM8K (Cobbe et al., 2021)), and instruction following data (Alpaca (Taori et al., 2023)) in terms of perplexity: we compute the average perplexity score of pre-trained LLMs over the concatenation of the question and ground-truth chain-of-thought response to examine how predictable the task is; the lower the perplexity is, the more predictable and the harder to learn the task is. We also include the perplexity over the pretraining corpus as a reference point.

Table 7: Single-task learning performance by evaluating the atomic CoT models on the in-domain atomic tasks.

| CoT Format | Next Letter EM | ASCII Mult EM | Concat EM | Skill-Mix Literary Full Marks | Skill-Mix Literary Skill Fraction | Skill-Mix Rhetorical Full Marks | Skill-Mix Rhetorical Skill Fraction |
|---|---|---|---|---|---|---|---|
| Llama 2-7B | | | | | | | |
| StandardCoT | 100.0 | 85.7 | 83.0 | 63.5 | 63.5 | 53.3 | 53.3 |
| ComposableCoT | 95.0 | 86.0 | 77.0 | 71.4 | 71.4 | 72.4 | 72.4 |
| Qwen 2.5-7B | | | | | | | |
| StandardCoT | 90.0 | 99.0 | 77.4 | 77.4 | 77.6 | 70.5 | 70.5 |
| ComposableCoT | 99.4 | 99.7 | 77.3 | 77.4 | 77.6 | 76.7 | 81.9 |

Table 8: Perplexity of the base models over the task datasets. For the string manipulation and arithmetic tasks, the perplexity score is averaged over the 3 atomic tasks and the 3 pairwise compositional tasks. Our evaluation datasets include text that is less predictable under pre-trained LLMs than other similar tasks.

| | Task Dataset | Pretraining WikiText | Math Reasoning GSM8K | Instruction Following Alpaca | String Manipulation And Arithmetic Avg. |
|---|---|---|---|---|---|
| Model | Llama 2-7B | 4.77 | 2.54 | 3.84 | 15.97 |
| | Qwen 2.5-7B | 5.93 | 2.38 | 4.68 | 7.02 |

Table 8 indicates that our selected tasks consist of text that is less typical under pre-trained language models than other similar tasks, particularly other popular reasoning tasks. The higher perplexity is likely due to these tasks requiring the model to operate on letters rather than words.

## G  TRAINING CONFIGURATIONS

### G.1  GENERAL CONFIGURATIONS

We conduct all fine-tuning experiments with LoRA(Hu et al., 2022) using the following set of hyperparameters: we use a rank of 8, $\alpha = 16$, and a dropout rate of 0.2 to prevent overfitting. We apply LoRA adapters to all linear modules, including the attention matrices $Q$, $K$, $V$ and MLP matrices of all layers. We use bfloat16 precision for training and we use the efficient implementation of LoRA by LlamaFactory (Zheng et al., 2024). We use a training batch size of 4 and train for 5 epochs for all experiments that share the same number of training data; for methods that use a potentially smaller amount of training data (e.g. RFT methods usually get fewer data examples than the number of compositional training data provided, depending on how many correct responses we can sample from the model), we adjust the batch size to match the number of steps.

### G.2  CONFIGURATION FOR REJECTION SAMPLING FINE-TUNING

In addition to the sampling parameters (see Section 4), we consider the following configuration of RFT for sampling the correct responses: if the model generates multiple correct responses for a given question, we only randomly select *one* of them to be added into the RFT dataset $D_{\mathrm{RFT}}$. In this way we ensures the diversity of examples in $D_{\mathrm{RFT}}$ so that the dataset will not be filled with samples from a small set of examples where the model is good at.

### G.3  HYPERPARAMETERS: LEARNING RATE

We find in preliminary experiments that learning rate is the most important hyperparameter for the fine-tuning experiments of our interest. We perform hyperparameter sweeps for learning rate over the space of $\{5e-3, 1e-3, 5e-4, 1e-4, 5e-5\}$ on a validation set for each experiment. The optimal learning rate for each method for the experiments with compositional supervision in Table 9.

Table 9: Optimal learning rate for each method in the experiments with compositional supervision.

| Category | Method | Next Letter + Mult | Concat + Next Letter | Concat + Mult | Skill-Mix Literary + Rhetorical |
|---|---|---|---|---|---|
| | | Llama 2-7B | | | |
| SFT | SFT on Base Model | 1e-3 | 1e-3 | 5e-4 | 5e-4 |
| | CFT on StandardCoT-Merge | 1e-3 | 5e-4 | 1e-4 | 1e-4 |
| | CFT on StandardCoT-MTL | 1e-4 | 1e-4 | 1e-4 | 1e-3 |
| MTL | StandardCoT + Comp Answer | 1e-3 | 5e-4 | 1e-3 | 5e-4 |
| RFT | StandardCoT-Merge | - | 1e-3 | 1e-3 | 5e-4 |
| | ComposableCoT-Merge (Ours) | 1e-4 | 1e-4 | 1e-3 | 1e-3 |
| | | Qwen 2.5-7B | | | |
| SFT | SFT on Base Model | - | 1e-3 | 1e-3 | 5e-4 |
| | CFT on StandardCoT-Merge | - | 5e-4 | 5e-4 | 1e-4 |
| | CFT on StandardCoT-MTL | - | 1e-3 | 1e-3 | 1e-3 |
| MTL | StandardCoT + Comp Answer | - | 5e-4 | 5e-4 | 1e-3 |
| RFT | StandardCoT-MTL | - | 1e-3 | 1e-4 | 5e-4 |
| | ComposableCoT-MTL (Ours) | - | 1e-3 | 1e-3 | 5e-4 |

Table 10: Data conditions for each task used for our evaluation.

| | | # Train | # Test |
|---|---|---|---|
| Atomic Tasks | Next Letter | 100 | 700 |
| | ASCII Mult | 100 | 700 |
| | Concat | 500 | 700 |
| | Skill-Mix Literary | 100 | 126 |
| | Skill-Mix Rhetorical | 100 | 105 |
| Compositional Tasks | Next Letter + Mult | 100 | 700 |
| | Concat + Next Letter | 100 | 504 |
| | Concat + Mult (Llama 2-7B) | 500 | 700 |
| | Concat + Mult (Qwen 2.5-7B) | 100 | 700 |
| | Skill-Mix Literary + Rhetorical | 100 | 245 |

### G.4 HYPERPARAMETERS: MODEL MERGING

For methods that use model merging as the combination, we use Task Arithmetic (Ilharco et al., 2023b) to combine the atomic CoT models. We perform a hyperparameter sweep for the scalars $\alpha$ and $\beta$ over the space of $\alpha \in \{0.1, 0.2, 0.3, 0.4, 0.5, 0.6, 0.7, 0.8, 0.9\}$ and $\beta = 1 - \alpha$ on a validation set for each task.

## H DATA STATISTICS

### H.1 GENERAL DATA CONDITIONS FOR EXPERIMENTS

Table 10 summarizes the number of training data and test data used in the evaluations in Sections 5.1 and 5.2. Note that for letter concatenation + multiplication we have two sizes of the compositional training data for Llama 2-7B and Qwen 2.5-7B: this is because all methods on Llama 2-7B perform poorly on zero-shot evaluation for this task and we need a slightly larger amount of compositional training data so that different methods can start to show distinguishable compositional task performance from each other. Regardless, we still consider 500 to be a reasonably small amount of training data, satisfying our ideal data conditions defined earlier.

Table 11: The detailed breakdown of the number of training data used by each method in the zero-shot setting. $N_i$ and $N_j$ denotes the number of training data from the atomic tasks $\mathcal{T}_i$ and $\mathcal{T}_j$ seen by the method during training.

|  | Method | $N_i$ | $N_j$ |
|---|---|---|---|
| Next Letter + Mult;
Skill-Mix Literary + Rhetorical | StandardCoT-Merge | 0 | 0 |
|  | ComposableCoT-Merge | 100 | 100 |
|  | StandardCoT-MTL | 100 | 100 |
|  | ComposableCoT-MTL | 100 | 100 |
| Concat + Next Letter;
Concat + Mult | StandardCoT-Merge | 500 | 100 |
|  | ComposableCoT-Merge | 500 | 100 |
|  | StandardCoT-MTL | 500 | 100 |
|  | ComposableCoT-MTL | 500 | 100 |

Table 12: The detailed breakdown of the number of training data used by each method with compositional supervision for Qwen 2.5-7B. $N_i$ and $N_j$ denotes the number of training data from the atomic tasks $\mathcal{T}_i$ and $\mathcal{T}_j$ seen by the method during training. $N_{(i,j)}$ denotes the number of compositional answer data seen during training.

|  | Method | $N_i$ | $N_j$ | $N_{(i,j)}$ |
|---|---|---|---|---|
| Next Letter + Mult;
Skill-Mix Literary + Rhetorical | SFT on Base Model | 0 | 0 | 100 |
|  | CFT on StandardCoT-Merge | 100 | 100 | 100 |
|  | CFT on StandardCoT-MTL | 100 | 100 | 100 |
|  | MTL on StandardCoT + Comp Answer | 100 | 100 | 100 |
|  | RFT on StandardCoT-Merge | 100 | 100 | 100 |
|  | RFT on ComposableCoT-Merge | 100 | 100 | 100 |
|  | RFT on StandardCoT-MTL | 100 | 100 | 100 |
|  | RFT on ComposableCoT-MTL | 100 | 100 | 100 |
| Concat + Next Letter;
Concat + Mult | SFT on Base Model | 0 | 0 | 100 |
|  | CFT on StandardCoT-Merge | 500 | 100 | 100 |
|  | CFT on StandardCoT-MTL | 500 | 100 | 100 |
|  | MTL on StandardCoT + Comp Answer | 500 | 100 | 100 |
|  | RFT on StandardCoT-Merge | 500 | 100 | 100 |
|  | RFT on ComposableCoT-Merge | 500 | 100 | 100 |
|  | RFT on StandardCoT-MTL | 500 | 100 | 100 |
|  | RFT on ComposableCoT-MTL | 500 | 100 | 100 |

## H.2 TRAINING DATA USED BY EACH METHOD

We show a detailed breakdown in Table 11 of the number of training data used by each zero-shot method for both models and in Table 12 for Qwen 2.5-7B by each method with compositional answer data in the experiments in Section 5.2. Note that the statistics for Llama 2-7B in the setting with compositional supervision are mostly the same except $N_{(i,j)} = 500$ for concat + next letter and concat + mult.

## I DETAILS OF THREE-WAY COMPOSITIONS

### I.1 DATA

We include 700 test examples for Letter Concat + Next Letter + Mult (*String Tasks*), and 245 test examples for Skill-Mix Literary + Rhetorical + Logical (*Skill-Mix*). For *Skill-Mix*, we additionally train an atomic model for Skill-Mix-Logical with 100 training examples.

Table 13: Summary of methods evaluated in the zero-shot compositional evaluation and the composition with limited compositional answer data. "Merge" stands for model merging; "MTL" stands for multitask learning; "CFT" stands for continued fine-tuning; "RFT" stands for rejection sampling fine-tuning. "-" means the property is not applicable to the method (e.g. *MTL on Standard + Comp Answer* mixes Standard CoT data with compositional answer data, and trains a single MTL model from the pretrained model, so there is no atomic CoT model trained or combined.)

| Method | # Atomic CoT Models Trained | Atomic CoT Format | Combination Method | Model trained on Compositional Data | How is Compositional Data Used |
|---|---|---|---|---|---|
| *Zero-shot Evaluation* | | | | | |
| StandardCoT-Merge | 2 | Standard | Merge | - | - |
| **ComposableCoT-Merge (Ours)** | 2 | Composable | Merge | - | - |
| StandardCoT-MTL | 1 | Standard | MTL | - | - |
| **ComposableCoT-MTL (Ours)** | 1 | Composable | MTL | - | - |
| *Evaluation with Limited Compositional Answer Data* | | | | | |
| CFT on StandardCoT-Merge | 2 | Standard | Merge | StandardCoT-Merge | CFT |
| CFT on StandardCoT-MTL | 1 | Standard | MTL | StandardCoT-MTL | CFT |
| MTL on StandardCoT + Comp Answer | - | Standard | - | Pretrained Model | Mix with Atomic CoT data and MTL |
| RFT on StandardCoT-Merge | 2 | Standard | Merge | StandardCoT-Merge | RFT |
| **RFT on ComposableCoT-Merge (Ours)** | 2 | Composable | Merge | ComposableCoT-Merge | RFT |
| RFT on StandardCoT-MTL | 1 | Standard | MTL | StandardCoT-MTL | RFT |
| **RFT on ComposableCoT-MTL (Ours)** | 1 | Composable | MTL | ComposableCoT-MTL | RFT |

## I.2 TRAINING AND INFERENCE

**Training**  We use the same data augmentation scheme to create atomic CoT training data as the one we use for two-way composition in Section 4. This means that we append only one proxy prefix to the prompt. The general scheme can insert at most $n-1$ proxy prefixes at the end of the prompt for $n > 2$, but we found that the test-time generalization scheme described in **Instantiation of Tags** under Section 3.1 works as well: adding only one proxy prefix achieves comparable compositional performance to adding two proxy prefixes while being more efficient during training, since the training data length is shorter. Thus, we experiment with the latter scheme.

**Inference**  We use the same inference strategy specified in **Data Construction** under Section 4: for zero-shot inference, we first sample a response from $M_{\text{comb}}$. Then, we repeat the following *twice*: we append *<suffix>* to the end of the generated response when it stops generation, and continue generation until the model stops again.

## J  FULL RESULTS FOR THE QUALITY ANALYSIS OF THE GENERATED CoTS

Table 14 includes the full results of the quality analysis of the generated CoTs using both multi-task learning (MTL) and model merging as the combination methods for atomic CoT models. Given the same combination method, combined Composable CoT models generate responses including both atomic CoT patterns more frequently than combined atomic CoT models.

## K  ERROR ANALYSES

In addition to not being able to perform the individual atomic task correctly, we show three types of common errors made by ComposableCoT variants in the zero-shot compositional evaluation setting.

1. Example K.1 shows an example where the generated CoT is only able to replicate CoT of one atomic CoT and repeat the same CoT in the prefix and suffix.

2. Example K.2 shows an example where the combined model fails to continue generation after generating the prefix CoT. This is a common error for Composable models combined with model merging.

Table 14: Intrinsic evaluation of the generated CoTs from atomic CoT models evaluated on the compositional task in the zero-shot setting. "% $\mathcal{T}_1$ CoT" denotes the percentage of generated responses that use the CoT format of the first atomic task of the composition, and likewise for the second. $\dagger$ denotes that the ComposableCoT method has a significantly higher "% Both CoT" than the StandardCoT counterpart at the 0.01 level using a paired bootstrap test. Combined Composable CoT models generate responses including both atomic CoT patterns more frequently than combined atomic CoT models.

|  | Method | Performance | % $\mathcal{T}_1$ CoT | % $\mathcal{T}_2$ CoT | % Both CoT |
|---|---|---|---|---|---|
| Next Letter + Mult | StandardCoT-Merge | 70.4 | 85.3 | 95.1 | 85.3 |
|  | ComposableCoT-Merge | 95.4 | 100.0 | 100.0 | $\dagger$**100.0** |
|  | StandardCoT-MTL | 3.6 | 0.0 | 100.0 | 0.0 |
|  | ComposableCoT-MTL | 96.3 | 98.9 | 100.0 | $\dagger$98.9 |
| Concat + Next Letter | StandardCoT-Merge | 77.0 | 90.3 | 98.7 | 90.0 |
|  | ComposableCoT-Merge | 75.4 | 91.6 | 100.0 | **91.6** |
|  | StandardCoT-MTL | 72.1 | 99.7 | 32.1 | 32.1 |
|  | ComposableCoT-MTL | 74.3 | 100.0 | 83.1 | $\dagger$81.3 |
| Concat + Mult | StandardCoT-Merge | 54.8 | 100.0 | 99.4 | **99.4** |
|  | ComposableCoT-Merge | 19.2 | 44.6 | 60.5 | 17.7 |
|  | StandardCoT-MTL | 60.9 | 100.0 | 66.7 | 66.7 |
|  | ComposableCoT-MTL | 63.3 | 100.0 | 85.9 | $\dagger$85.0 |
| Skill-Mix Literary + Rhetorical | StandardCoT-Merge | 29.8 | 60.0 | 59.2 | 35.9 |
|  | ComposableCoT-Merge | 39.6 | 64.1 | 66.9 | $\dagger$**43.3** |
|  | StandardCoT-MTL | 42.0 | 65.3 | 58.0 | 37.6 |
|  | ComposableCoT-MTL | 49.0 | 64.5 | 65.7 | $\dagger$42.0 |

3. Example K.3 shows an example where the combined model uses the wrong atomic CoT in the prefix that should have been used in the suffix.

---

**K.1 Error Case: Replicating One Atomic CoT Pattern**

```
[Instruction]
Take the first letter of each word in the sequence, concatenate
them in lower case, and find the next letter in alphabet
following the last letter in the sequence of the concatenated
letters:  Jessika Berk Hamilton Stephanie answer:
[Answer String]
t
[Model Output]
<prefix> The last letter is e, and the letter following it in
alphabet is f.  So the answer is f.</prefix> <suffix> The last
letter is e, and the letter following it in alphabet is f.  So
the answer is f.</suffix>
[Predicted Answer]
f
```

---

**K.2 Error Case: Failing to Continue Generation after Prefix CoT**

```
[Instruction]
Take the last letter of each word in the sequence, concatenate
them in lower case, then find the ASCII value of the last letter
in the sequence of the concatenated letters, and multiply the
ASCII value by 6:  Ayat Amaris Heavenly Mariela answer:
[Answer String]
582
```

```
[Model Output]
<prefix> The last letter of the 1st word is t.  The last letter
of the 2nd word is s.  The last letter of the 3rd word is y.
The last letter of the 4th word is a.  So the answer is tasy,
and the ASCII value of the last letter in the sequence of the
concatenated letters is 121, so the answer is 726.</prefix>
<suffix>
[Predicted Answer]
726
```

### K.3 Error Case: Use the Wrong Order of Atomic CoTs

```
[Instruction]
Take the second letter of each word in the sequence, concatenate
them in lower case, and find the next letter in alphabet
following the last letter in the sequence of the concatenated
letters:  D'shaun Takia Hiroki Arjun answer:
[Answer String]
s
[Model Output]
<prefix> the last letter is n, and the letter following it in
alphabet is o.  so the answer is o.</prefix> <suffix> the second
letter of d is d, the second letter of t is a, the second letter
of h is i, and the second letter of a is r.  so the answer is
dair.</suffix>
[Predicted Answer]
dair
```

## L   COMPUTATION RESOURCES

We conducted all our experiments primarily on a server with NVIDIA RTX A6000 48GB GPU, and a server with GPU of NVIDIA GH200 120GB. Each individual experiment can be run on a single A6000 of 48 GB within 1 hour.

## M   LICENSING

We use the following publicly available datasets from prior works with open licenses.

**Skill-Mix**   Yu et al. (2024) uses the Apache-2.0 license and data is available at: `https://huggingface.co/spaces/dingliyu/skillmix`.

**Letter concatenation**   The dataset uses the Apache-2.0 license and the data is available at: `https://huggingface.co/datasets/ChilleD/LastLetterConcat`

