# OpenReview forum: "Learning Composable Chains-of-Thought"
_ICLR.cc/2026/Conference — ICLR 2026 Conference Withdrawn Submission_

### Official Review · Reviewer_ttWX · 2025-10-27

**Soundness:** 3
**Presentation:** 3
**Contribution:** 4
**Rating:** 6
**Confidence:** 4

**Summary:**

This paper introduces a method for enabling LLMs to generalize to compositional reasoning tasks without using compositional chain-of-thought (CoT) data. By augmenting CoTs of atomic tasks with composable prefixes, the model learns to combine reasoning steps across tasks. Experiments across natural language, string manipulation, and arithmetic show that training on Composable CoT data outperforms multitask and fine-tuning baselines under the same data budget.

**Strengths:**

- This study conducts an interesting and important research topic: generalization to compositional reasoning tasks by using only atomic reasoning data at training.
- The experiments demonstrate the effectiveness of the proposed method though they are toy experiments.
- The paper is well-written and easy to follow.

**Weaknesses:**

- I think we need an additional ablation study about why the simple trick (adding just random tags) works. For example, which is more important, tag or random text? what if we remove or change the tag? what if we change the style of random text?

**Questions:**

- Table1: Does the result not apply RFT(rejection sampling fine-tuning)? I think there is no such description, but based on the context I feel so.

---

> ### Author Response · Authors · 2025-11-21
>
> We thank the reviewer for the feedback. We will address some of the comments here.
>
> > An additional ablation study about why the tags work: which is more important, the tag or the random prefix? What if we remove or change the tag? What if we change the style of random text?
>
> We perform an ablation on the use of tags to demonstrate its importance. We include different variants of composable CoT with Qwen2.5-7B on zero-shot compositions: (a) Composable CoT - Prefix/Suffix (Default): The method we present in the paper that uses random proxy prefixes and the \<prefix> \<suffix> tags; (b) Composable CoT - Think: We use random proxy prefixes, but change the tags from \<prefix>\<suffix> to \<think>; (c) Composable CoT - No Tag: We use random proxy prefixes, but remove the \<prefix> \<suffix> tags.  We use multi-task learning for all ComposableCoT variants.
>
> | Method                                   | Use Proxy Prefix? | Use Tag?               | Next Letter + Mult | Concat + Next Letter | Concat + Mult | Skill-Mix  | |
> | ---------------------------------------- | ----------------- | ---------------------- | ------------------ | -------------------- | ------------- | ---------- | --------- |
> |                                          |                   |                        | Exact Match                 | Exact Match                   | Exact Match            | Full Marks | Skill Fraction |
> | Standard CoT                             | No                | No                     | 3.6                | 60.9                 | 72.1          | 42.0       | 58.2 |
> | Composable CoT - Prefix/Suffix (Default) | Yes               | Yes, \<prefix>/\<suffix> | 96.3               | 63.3                 | 74.3          | 49.0       | 66.7 |
> | Composable CoT - Think                   | Yes               | Yes, \<think>           | 47.6               | 20.4                 | 76.7          | 52.7       | 68.8 |
> | Composable CoT - No Tag                  | Yes               | No                     | 34.6               | 62.1                 | 70.4          | 48.2       | 66.0 |
>
>
>
> In the table above, we observe that tags are important: for methods that use proxy prefixes (Composable CoT - Prefix/Suffix and Composable CoT - Think versus Composable CoT - No Tag), **using tags leads to better performance across tasks**. In addition, **the specific instantiations of tags matter less on the performance**. For methods that use different tags (Composable CoT - Prefix/Suffix versus Composable CoT - Think), the results are mixed, and no tag leads to consistently better performance than the other tag for all tasks.
>
>
> > In Table 1, does the result not apply rejection sampling fine-tuning (RFT)?
>
> We indicate that in Section 4 and Section 5.1 that Table 1 shows results of zero-shot composition **without compositional supervision**, which means that we do not apply RFT. We will clarify this in a future version of the paper.

---

### Official Review · Reviewer_2jkA · 2025-11-01

**Soundness:** 2
**Presentation:** 2
**Contribution:** 2
**Rating:** 2
**Confidence:** 4

**Summary:**

This paper proposes a data augmentation method that adds random prefixes to atomic CoT steps to improve compositional generalization. The idea is evaluated on synthetic reasoning tasks such as string manipulation, arithmetic, and Skill-Mix.

**Strengths:**

The proposed method is simple and shows significant improvements on synthetic tasks.

**Weaknesses:**

1. The paper only conducts experiments on synthetic tasks. It is unclear how the proposed method can be applied to real-world scenarios such as math reasoning or code generation. In particular, identifying the atomic tasks in these domains is non-trivial. I also doubt whether simply data augmentation without explicitly training the model for composition can lead to meaningful improvements on realistic tasks.
2. I am uncertain about the broader impact of this work. To advance the frontier of model's reasoning capabilities, the current standard practice is to use reinforcement learning. When compute is limited,  distillation from stronger LLMs also achieve good result. I do not see clear evidence that the proposed method would bring better results on real-world tasks compared to these existing approaches. It would be helpful if the authors could discuss how their idea might be applied in more realistic training setups or combined with current methods.
3. The discussion of related work is insufficient. The authors should at least include: (1) recent methods for training models to generate long CoTs, such as distillation , self-training [1], or reinforcement learning[2]; and (2) broader discussions on understanding and improving compositional generalization in LLMs.

[1] Zelikman et al. STaR: Bootstrapping Reasoning With Reasoning

[2] Guo et al. DeepSeek-R1: Incentivizing Reasoning Capability in LLMs via Reinforcement Learning

**Questions:**

1. Why is the performance of SFT with compositional supervision so low on the first three tasks (Table 1)? It would be helpful if the authors could provide more details on the setup of this baseline, including examples of training samples.
2. It is somewhat surprising that augmenting data with random prefixes improves model performance, given the significant distribution shift between training and testing. Do the authors have an explanation for that? In addition, I am curious whether such training leads to a larger degradation on other capabilities (e.g., instruction following) compared to other baselines.
3. For ComposableCoT-Merge, how is the scaling factor determined?

---

> ### Author Response · Authors · 2025-11-21
>
> Thanks to the reviewer for the comments! We address the questions below.
>
> > The paper only conducts experiments on synthetic tasks. It is unclear how the proposed method can be applied to real-world scenarios such as math reasoning or code generation. In particular, identifying the atomic tasks in these domains is non-trivial. I also doubt whether simply data augmentation without explicitly training the model for composition can lead to meaningful improvements on realistic tasks.
>
> We intentionally include both synthetic tasks for experimental controllability and a more realistic task (Skill-Mix). Skill-Mix does capture aspects of real-world language understanding: models are required to write sentences using writing techniques, similar to what might happen in a co-writing model. Since we have observed consistent improvements led by using ComposableCoT on such tasks, we believe there is evidence that our data augmentation method works for real-world tasks, including but not limited to creative writing.
>
> Moreover, from the perspective of controlled experiments, we note that our task design has several advantages over other tasks, such as math reasoning or code generation, as follows.
>
> (1) **Atomic skills are distinguishable**: To evaluate compositional generalization, the atomic skills need to be distinguished from each other so that learning one skill is independent from learning another skill. Our task design aims at making the distinguishability as strong as possible.
>
> (2) **Compositions are unseen**: Given the size of the corpus LLMs are pre-trained on, it is hard to ensure that compositional examples drawn from general reasoning tasks, such as math word problems, are not similar to the data seen during pre-training. Our task designs aim at creating compositional examples that are less commonly seen in the pre-training data.
>
> In addition, we note that, as the reviewer has also indicated, identifying atomic tasks in more complex domains is non-trivial. For example, [1] identifies 4 atomic skills that are important for math reasoning, but such skills are not consistently helpful for all models and in all settings within the same domain. Thus, we leave it for future work, as our goal is a scientific understanding of compositional generalization of LLMs. Our method should be generalizable to such domains once a well-defined set of atomic tasks is identified.
>
> > I am uncertain about the broader impact of this work. To advance the frontier of model's reasoning capabilities, the current standard practice is to use reinforcement learning. When compute is limited, distillation from stronger LLMs also achieve good result. I do not see clear evidence that the proposed method would bring better results on real-world tasks compared to these existing approaches. It would be helpful if the authors could discuss how their idea might be applied in more realistic training setups or combined with current methods.
>
> There is a potential misunderstanding here, since we believe we have addressed this point already. First, the rejection sampling fine-tuning (RFT) algorithm we used to improve composition in Section 3 is a form of reinforcement learning, as shown in [2] [3]. Second, **our data augmentation approach is complementary to other post-training algorithms, such as RLVR**. The fact that RFT on composable CoT models outperforms RFT on standard CoT models indicates that our method can be further strengthened when used in combination with RL methods. In addition to RFT, other RL algorithms such as GRPO can potentially fit into our self-boostrapping pipeline and be combined with composable CoT; we leave this for future work.

---

> ### Author Response · Authors · 2025-11-21
>
> > The discussion of related work is insufficient. The authors should at least include: (1) recent methods for training models to generate long CoTs, such as distillation, self-training, or reinforcement learning; and (2) broader discussions on understanding and improving compositional generalization in LLMs.
>
> We would like to clarify the following:
>
> (1) **Our method is a general data augmentation method that is agnostic to the length of SFT traces.** Although we have only evaluated composable CoT on short CoT training, our method can be potentially applied to long CoT training as well, using the same augmentation strategy. Given the generality of our method beyond the length of CoT, we do not think it is appropriate to include specific discussions on the literature of test-time scaling.
>
> (2) **We have included an in-depth discussion on different distillation methods from the literature in both related work and in our baselines.** Algorithm-wise, in Table 2, we compared our method with distillation algorithms, including standard SFT, continued fine-tuning, and multi-task SFT. Data-wise, we included both teacher-student distillation (e.g., all Skill-Mix experiments use atomic CoTs distilled from GPT-4o) and self-training distillation (e.g., rejection sampling fine-tuning used in Section 3.3). We note that our RFT approach is very similar to the method of STaR [2]; we will highlight this conclusion more strongly in any future version.
>
> (3) **Our paper has already engaged with the mainstream paradigms for improving compositional generalization from the literature** by directly comparing against the representative baselines: (a) In-context learning [4, 5]: represented by the few-shot prompting baseline in Table 1; (b) Curriculum learning [6, 7]: represented by the continued fine-tuning baseline in Table 2. In addition, as discussed in the Introduction, our method is also grounded in the understanding of compositional generalization in LLMs from the literature that the compositional reasoning capability of LLMs can be improved by generating CoT [8, 9]. We will include an extended version of the related work to emphasize the discussion of understanding and improving compositional generalization in LLMs in a future version of the paper.
>
> > Why is the performance of SFT with compositional supervision so low on the first three tasks (Table 1)? It would be helpful if the authors could provide more details on the setup of this baseline, including examples of training samples.
>
> The SFT with compositional supervision baseline in Table 1 only uses compositional training data with the answer only (rather than with CoT), while other methods in Table 1 use CoT. This explains the low performance of this baseline, as it does not utilize CoT for reasoning. We have already included the setup of this baseline in Section 4 (lines 318 - 319) and training examples in Appendix C and D; we will clarify this more in a future version of the paper.
>
> > It is somewhat surprising that augmenting data with random prefixes improves model performance, given the significant distribution shift between training and testing. Do the authors have an explanation for that?
>
> We hypothesize that using random prefixes as prefixes is more robust to out-of-domain CoTs, while training on other variants of proxy prefixes, including random sentences and random parts of the prompt, might overfit the training distribution of prefix CoTs. Moreover, using random prefixes ensures that we do not need to have any prior knowledge about the structure/content of the proxy CoT prefixes to train atomic models with our method for compositional generalization. This ensures rigorous evaluation of compositional generalization.

---

> ### Author Response · Authors · 2025-11-21
>
> > In addition, I am curious whether such training leads to a larger degradation on other capabilities (e.g., instruction following) compared to other baselines.
>
> We evaluate the Qwen2.5-7B models fine-tuned on atomic tasks of string manipulation and arithmetics on out-of-domain tasks. Note that the base models we used in the paper are not instruction-tuned models, so we chose two representative out-of-domain tasks, Winogrande and MMLU, to evaluate the general capabilities of LLMs. We use 5-shot prompting, following the same protocol in [10].
>
> |                      |                 | In-Domain             | Out-of-Domain | |
> | -------------------- | --------------- | --------------------- | ------------- | ------------- |
> | Training Data        | Model           | Zero-shot Composition | Winogrande    | MMLU |
> |                      | Base            | \-                    | 76.3          | 74.2 |
> | Next Letter + Mult   | StandardCoT-MTL | 96.3                  | 72.1          | 72.1 |
> | | ComposableCoT-MTL    | 3.6             | 75.4                  | 74.1          |
> | Concat + Next Letter | StandardCoT-MTL | 60.9                  | 69.9          | 64.9 |
> | | ComposableCoT-MTL    | 63.3            | 71.0                  | 66.0          |
> | Concat + Mult        | StandardCoT-MTL | 72.1                  | 75.0          | 73.7 |
> | | ComposableCoT-MTL    | 74.3            | 75.5                  | 74.2          |
>
>
> The table shows that although fine-tuning on atomic tasks leads to degradation on OOD tasks compared with the base model, **ComposableCoT models have smaller OOD degradation than StandardCoT models.** This indicates that training on ComposableCoT with random prefixes mitigates forgetting better than training on StandardCoT.
>
> > For ComposableCoT-Merge, how is the scaling factor determined?
>
> As shown in Appendix G.4, we perform a hyperparameter sweep on a validation set of each task.
>
> [1] Gandhi et al., 2025. Cognitive Behaviors that Enable Self-Improving Reasoners, or, Four Habits of Highly Effective STaRs.
>
> [2] Zelikman et al., 2022. STaR: Bootstrapping Reasoning With Reasoning.
>
> [3] Chang et al., 2024. RL-STaR: Theoretical Analysis of Reinforcement Learning Frameworks for Self-Taught Reasoner.
>
> [4] Levy et al., 2023. Diverse demonstrations improve in-context compositional generalization.
>
> [5] Chen et al., 2024. Skills-in-context: Unlocking compositionality in large language models.
>
> [6] Hase et al., 2024. The unreasonable effectiveness of easy training data for hard tasks.
>
> [7] Sun et al., 2024. Easy-to-hard generalization: Scalable alignment beyond human supervision.
>
> [8] Li et al., 2024. Chain of thought empowers transformers to solve inherently serial problems.
>
> [9] Li et al., 2023. Dissecting chain-of-thought: Compositionality through in-context filtering and learning.
>
> [10] Yang et al., 2024. Qwen2.5 Technical Report.

---

### Official Review · Reviewer_KsW6 · 2025-11-01

**Soundness:** 3
**Presentation:** 3
**Contribution:** 2
**Rating:** 4
**Confidence:** 4

**Summary:**

This paper claims a way to make chain-of-thought reasoning "composable": train on atomic tasks, add random "proxy prefix" strings in front of the CoT, tag them, then fine-tune the LLM so it learns to generate composable CoTs conditioned on the prefix. At test-time, there is a  concatenate strategy at the task level (via multitask learning or model-merging). On simple synthetic tasks: string ops, ASCII arithmetic, toy natural-language skills, the authors show often big improvements over the non-composable CoT strategy but only at the level of 2 compositions. Improvements wrt control baselines for 3 compositions is also observed, also to a lesser extent

**Strengths:**

1. The problem of compositional generalization is a core ML problem and I appreciate the authors trying to address it in the modern setting: standard LLMs have been shown to be incapable of large scale compositional reasoning. This approach is interesting and tries to leverage CoT that has been shown to help with logical/math reasoning for compositional generalization.
2. Well written and motivated empirically.
3. Strong results of the proposed approach over baselines are interesting to see, though notably janky at many places especially table 1 llama 2.

**Weaknesses:**

1. The main weakness of this paper is that it doesn't experiment with compositionality enough. The tasks are also not compositional enough and there is a risk of templating/pattern-matching hacking going on here. Symbolic manipulation of some task with quantifiable controllable compositionality (e.g. n digit multiplication. Multiplication of n digits k times) would be interesting to see. 2 way compositional results are interesting to study as a starter but you should not stop at 3 way compositions. What about k-way compositions? I already see the improvement delta declining at 3 compositions.
2. Results for small models have high variance. Table 1 llama and qwen 7b have a huge performance delta. The benefits of the proposed approach are unclear, especially in light of 1.
3. Results on larger models are missing. Presumably compositional CoTs should yield even better performance deltas for larger models e.g 30b, 70b?
4. Proxy prefixes can also be interpreted as random noise and this approach feels rather like prompt regularization. The experimental results again don't investigate the compositional utility of the approach.
5. Just using thinking tokens ala., Deepseek R1 distillation, and seeing performance effects is unexplored. This is also a clear baseline.

**Questions:**

see above

---

> ### Author Response · Authors · 2025-11-21
>
> We appreciate the comments from the reviewer, and we would like to address them below.
>
> > The paper doesn't experiment with compositionality enough, and the tasks are not compositional enough. There is a risk of templating/pattern-matching hacking.
>
> We argue that **controlled compositions evaluated in our paper are necessary for rigorous evaluation of compositional generalization with LLMs** for the following reasons.
>
> (1) **Atomic skills are distinguishable**: To evaluate compositional generalization, the atomic skills need to be distinguished from each other so that learning one skill is independent from learning another skill. Our task design aims at making the distinguishability as strong as possible.
>
> (2) **Compositions are unseen**: Given the size of the corpus LLMs are pre-trained on, it is hard to ensure that compositional examples drawn from general reasoning tasks, such as math word problems, are not similar to the data seen during pre-training. Our task designs aim at creating compositional examples that are less commonly seen in the pre-training data.
>
> We note that our task design satisfies the above considerations for experimenting with compositionality. Moreover, our evaluation setting is not only limited to controlled compositions of string manipulation and arithmetic, but also includes a creative writing task (Skill-Mix) that captures aspects of real-world language understanding: models are required to write sentences using writing techniques, similar to what might happen in a co-writing model. Given this is a free-form writing task, there is little room for templating and pattern hacking by design. Therefore, we believe the suite of evaluation tasks used in this paper is representative enough for scientific understanding of compositional generalization of LLMs.
>
>
> > Symbolic manipulation of some task with quantifiable controllable compositionality (e.g., n-digit multiplication. Multiplication of n digits k times) would be interesting to see.
>
> We appreciate the suggestions from the reviewer. However, we consider length generalization, the phenomenon represented by the n-digit multiplication and k-time multiplication of n-digit numbers, out of scope for the focus of our paper. As defined in [1], compositionality has three components: systematicity, productivity, and primitive application. **Our work mainly focuses on systematicity, the capability of applying known components (atomic skills) in unseen combinations (compositional skills).** Length generalization, on the contrary, refers to the productivity aspect, the capability of generalization to longer sequences of combinations than the combinations seen in training (e.g., from 2-digit multiplication to 6-digit multiplication); this is different from systematic compositionality because our setting does not assume that we have seen CoTs being composed at training time. Since these two types of compositionality require very different capabilities, we leave length generalization and related experiments for future work.
>
> > 2-way compositional results are interesting to study as a starter, but you should not stop at 3-way compositions. What about k-way compositions?
>
> We believe that the 2-way and 3-way composition results shown in the paper already demonstrate the potential of our method generalizing to k-way compositions of higher k. We note that constructing a high-quality evaluation suite for k-way composition of a larger k is nontrivial, and we leave it for future work.
>
> > Results for small models have high variance. Llama and Qwen 7b results in Table 1 have a huge performance delta. The benefits of the proposed approach are unclear.
>
> The performance delta between Llama-2-7B and Qwen2.5-7B in Table 1 is expected, given that Qwen2.5 models are stronger than Llama-2 models in terms of the pre-trained capabilities (as shown by mainstream benchmark results). Despite that, performance gains by our method over the baseline methods are consistent across different models and different tasks.

---

> ### Author Response · Authors · 2025-11-21
>
> > Results on larger models are missing.
>
> We evaluated Qwen2.5-14B-base on the SkillMix benchmark since ComposableCoT already achieves good performance on other benchmarks with the 7B-scale models. We compared ComposableCoT models with StandardCoT models, both using multi-task learning (MTL) as the combination method, for zero-shot 2-way composition and 3-way composition, following the same protocol as specified in Section 4 and Section 6.1. Results in the table below show that **at the 14B scale, models trained with Composable CoT still outperform models trained with StandardCoT.** This demonstrates that our method can potentially be applied to larger models as well, though we leave the evaluations on even larger models for future work, given the limits of computational resources.
>
> |                | Evaluation Metrics|StandardCoT-MTL-14B | ComposableCoT-MTL-14B (Ours) |
> | -------------- | ------------------- |  ------------------- | --------------------- |
> | SkillMix 2-way | Full Mark           | 60.8                  | **67.8** |
> | | Skill Fraction | 73.6                | **78.6**                  |
> | SkillMix 3-way | Full Mark           | 47.8                  | **52.2** |
> | |Skill Fraction | 71.8                | **74.7**                  |
>
>
>
>
>
> > Proxy prefixes can also be interpreted as random noise and this approach feels rather like prompt regularization. The experimental results again don't investigate the compositional utility of the approach.
>
> We are not completely sure what the reviewer refers to with the term “prompt regularization.” We argue that our approach is **prompt augmentation** that teaches LLMs to robustly combine and compose different CoT traces of atomic skills at inference time. The random proxy prefixes seen at train time ensure that the models do not overfit a specific distribution of prefix CoTs and make the models robust to any arbitrary prefix CoT encountered at inference time.
>
> Moreover, we analyze the compositional utility of our approach in Section 6.2: we’ve shown that models trained on composable CoT generate more responses that utilize the composition of atomic CoT traces, while models trained on standard CoT generate fewer compositions.

---

> ### Author Response · Authors · 2025-11-21
>
> > A clear baseline is R1-style distillation and/or with thinking tokens. (e.g., deepseek-R1)
>
> Our method is a general data augmentation method that is **agnostic to the length of SFT traces** and thus **orthogonal** to R1-style long CoT distillation. Although we have only evaluated composable CoT on short CoT training, our method can be potentially applied to long CoT training as well using the same augmentation strategy. Given the generality of our method beyond the length of CoT, **we do not think distilling long CoT traces from R1 into smaller models is an apples-to-apples fair baseline to ComposableCoT.**
>
> Regardless, we compare ComposableCoT-MTL using short CoT data (the setting in our paper) with R1-style distillation methods using long CoT data below. We evaluate two variants of R1 distillation: (a) General long CoT distillation (R1-distill-general): We use the DeepSeek-R1-Distill-Qwen-7B model from [2], which has been fine-tuned on 800k long CoT samples generated by DeepSeek-R1 on general reasoning tasks. This model is able to generate long CoT for reasoning, but is not trained on in-domain atomic skills. (b) In-domain long atomic CoT distillation (R1-distill-atomic): We distill long CoT traces by prompting Deepseek-R1 on atomic tasks and collect correct responses as the SFT data for Qwen2.5-7B-base model. We use the same protocol as specified in Section 4. This model is able to generate long CoT and is also trained on in-domain atomic skills.
>
> |                            | Next Letter + Mult     | Concat + Next Letter | Concat + Mult | Skill-Mix  | |
> | -------------------------- | ---------------------- | -------------------- | ------------- | ---------- | ---------- |
> |                            | Performance (↑)        |                      |
> |                            | EM                     | EM                   | EM            | Full Marks | Skill Fraction |
> | R1-Distill-General         | 50.9                   | 71.7                 | 63.7          | 19.6       | 24.6 |
> | R1-Distill-Atomic          | 76.9                   | **95.0**                   | **92.0**            | 33.5       | 39.6 |
> | ComposableCoT-MTL-ShortCoT | **96.3**                   | 63.3                 | 74.3          | **49.0**         | **66.7** |
> |                            | Avg. Output Tokens (↓) |
> | R1-Distill-General         | 422.3                  | 1081.4               | 782.6         | 555.4      |  |
> | R1-Distill-Atomic          | 3956.8                 | 2264.0               | 2090.9        | 1277.5     |  |
> | ComposableCoT-MTL-ShortCoT | 112.2                  | 102.9                | 98.7          | 357.2      |
>
>
> The table above shows that training models on long CoT of the atomic tasks can achieve strong performance on Concat + Next Letter and Concat + Mult, while using 8x - 20x more output tokens than our method trained on short CoT. On these tasks, augmenting R1 traces with ComposableCoT format can potentially further improve the performance. However, on Skill-Mix and Next Letter + Mult, R1-distill methods using long CoT are worse than ComposableCoT-MTL using short CoT, while using 2x - 20x more output tokens than our method trained on short CoT. **This indicates that simply training on long CoT without any formatting augmentation does not solve these compositional tasks by default.** In light of such findings, we emphasize that our length-agnostic augmentation method is important for improving compositional generalization in LLMs.
>
> [1] Fodor and Pylyshyn, 1988. Connectionism and cognitive architecture: A critical analysis.
>
> [2] Deepseek-AI et al., 2025. DeepSeek-R1: Incentivizing Reasoning Capability in LLMs via Reinforcement Learning.

---

> > ### Comment · Reviewer_KsW6 · 2025-11-27
> > **Thanks for the follow up! I still maintain that systematic compositionality necessitates k compositions with k > 5-10 for this technique to be really useful for the broader community**
> >
> > Thanks for the follow up! I still maintain that systematic compositionality necessitates k compositions with k > 5-10 for this technique to be really useful for the broader community cf. this [paper](https://arxiv.org/pdf/1711.00350)...
> >
> > This is a fairly standard ask in the systematic generalization community. These are still unseen in a "combinatorial" sense.

---

### Official Review · Reviewer_VYUV · 2025-11-02

**Soundness:** 2
**Presentation:** 2
**Contribution:** 2
**Rating:** 4
**Confidence:** 3

**Summary:**

The paper proposes Composable CoT, a lightweight data augmentation that wraps atomic CoT traces with tags and inserts “proxy prefixes” (random-letter strings) so that training examples look like “a CoT conditioned on previous CoTs.” At test time, atomic CoT models are combined either via multitask learning (MTL) or Task Arithmetic–style model merging; limited compositional supervision is optionally added through rejection-sampling fine‑tuning (RFT). On synthetic string/arithmetic compositions and on Skill‑Mix (literary+rhetorical skills), Composable CoT improves zero‑shot and small‑shot composition over standard CoT training, with the largest reported gains coming from RFT on top of the composable formatting.

**Strengths:**

Turning atomic CoT data into a “composable” format is easy to implement (two tags + random‑letter prefixes) and consistently lifts zero‑shot/limited‑shot composition across tasks and two 7B bases. The construction is clearly depicted (Fig. 2), and ablations show random letters are the most robust proxy prefix out‑of‑domain.

**Weaknesses:**

1. "Zero-shot" claim is fragile due to validation-time merging sweeps. For Task Arithmetic, the paper sweeps $\alpha, \beta$ on a validation set for each task (App. G.4). If this validation set is the compositional task, tuning leaks target supervision into model selection and weakens the zero-shot claim; at minimum this needs to be clarified and a version without compositional validation should be reported.
2. Heavy reliance on explicit tags and random prefixes; external validity is limited. The method depends on <prefix>/<suffix> markers and training on random-letter prefixes. This encourages learning a format protocol rather than discovering composition in untagged, natural inputs. The paper itself instantiates all intermediate tags as <prefix> and the final as <suffix> (Instantiation of Tags), underscoring the dependence on explicit scaffolding. Results without any tags or with naturally occurring preceding CoTs are not shown.
3. Evidence of reading and re-using intermediate results is weak. The "quality" analysis checks whether both atomic CoT templates appear, not whether the model actually consumes the earlier step's outputs (step-level causal dependence or variable-passing). This leaves open the possibility that the model just regenerates both CoTs in the suffix.
4. Many findings are unsurprising and the novelty is modest. The main lift plausibly comes from making multi‑CoT sequences in‑distribution via formatting. The work’s practical impact beyond controlled compositions may be limited without evidence on untagged, naturally compositional tasks (e.g., program‑of‑thought planning, tool‑use pipelines).

**Questions:**

See weaknesses

---

> ### Author Response · Authors · 2025-11-21
>
> We appreciate the comments from the reviewer, and we will address the questions as follows.
>
> > The method depends on the usage of [object Object]/[object Object] markers and random prefixes and requires explicit scaffolding. Results without any tags or with naturally occurring preceding CoTs are not shown.
>
> First, we clarify that we did not use “ [object Object]/[object Object] markers” in our proposed method. Rather, we use general-purpose tags, which we instantiate as \<prefix> and \<suffix> for two-way compositions. We argue that the usage of explicit tagging and conditioning on random prefixes does not affect the generality of our proposed method, and it rather empowers the trained models with robust compositional generalization, for the following two reasons.
>
> (1) In our method, tags are only used to separate different semantic components in the response, and they are not restricted to any specific instantiations. **Using tags as general-purpose semantic separators is a common practice in LLM post-training literature**: for example, RLVR [1,2] commonly uses \<think> tags to separate CoT traces from answers, and tool-use literature [3,4] uses \<tool> tags to separate tool outputs from other components. Moreover, the literature has shown that tags do not affect the generality of trained models, and without having tags, reasoning post-training might become unstable.[5] **Our training pipeline also designs tags to be general-purpose separators of different CoTs, so any instantiation would work as well as the specific \<prefix> \<suffix> instantiation we used.** Adding tags also helps the models to learn to scaffold and compose different CoTs instead of using only one atomic skill:  we’ve shown empirical evidence of this in Section 6.2. Thus, using explicit tags is a better design choice based on the literature and our empirical observations.
>
> We perform an ablation on the use of tags to demonstrate its importance. We include different variants of composable CoT with Qwen2.5-7B on zero-shot compositions: (a) Composable CoT - Prefix/Suffix (Default): The method we present in the paper that uses random proxy prefixes and the \<prefix> \<suffix> tags; (b) Composable CoT - Think: We use random proxy prefixes, but change the tags from \<prefix>\<suffix> to \<think>; (c) Composable CoT - No Tag: We use random proxy prefixes, but remove the \<prefix> \<suffix> tags.  We use multi-task learning for all ComposableCoT variants.
>
> | Method                                   | Use Proxy Prefix? | Use Tag?               | Next Letter + Mult | Concat + Next Letter | Concat + Mult | Skill-Mix  | |
> | ---------------------------------------- | ----------------- | ---------------------- | ------------------ | -------------------- | ------------- | ---------- | --------- |
> |                                          |                   |                        | Exact Match                 | Exact Match                   | Exact Match            | Full Marks | Skill Fraction |
> | Standard CoT                             | No                | No                     | 3.6                | 60.9                 | 72.1          | 42.0       | 58.2 |
> | Composable CoT - Prefix/Suffix (Default) | Yes               | Yes, \<prefix>/\<suffix> | 96.3               | 63.3                 | 74.3          | 49.0       | 66.7 |
> | Composable CoT - Think                   | Yes               | Yes, \<think>           | 47.6               | 20.4                 | 76.7          | 52.7       | 68.8 |
> | Composable CoT - No Tag                  | Yes               | No                     | 34.6               | 62.1                 | 70.4          | 48.2       | 66.0 |
>
>
>
> In the table above, we observe that tags are important: for methods that use proxy prefixes (Composable CoT - Prefix/Suffix and Composable CoT - Think versus Composable CoT - No Tag), **using tags leads to better performance across tasks**. In addition, **the specific instantiations of tags matter less on the performance**. For methods that use different tags (Composable CoT - Prefix/Suffix versus Composable CoT - Think), the results are mixed, and no tag leads to consistently better performance than the other tag for all tasks.
>
>
> (2) **Using random prefixes is more robust than using naturally occurring prefixes.** We include ablations of proxy prefix construction in Appendix B in our paper: using random letters as prefixes is more robust to out-of-domain CoTs than other variants of proxy prefixes, including random sentences and random parts of the prompt. Random prefix ensures that we do not need to have any prior knowledge about the structure/content of the proxy CoT prefixes to train atomic models with our method for compositional generalization.

---

> > ### Author Response · Authors · 2025-11-21
> >
> > > In the zero-shot composition setting with model merging, hyperparameter sweeping on compositional validation data potentially leaks target supervision into model selection and weakens the zero-shot claim. A version without compositional validation should be reported.
> >
> > First, we emphasize that all models have been fairly compared in this setting. We performed sweeping for the hyperparameters of the scaling factors on the validation set of compositional data for all merging-based methods, including Composable CoT + Merging (ours) and Standard CoT + Merging (baseline). Since the same model selection method is applied to our method and baselines, sweeping should not affect the relative performance ranking across methods in the zero-shot composition setting.
> >
> > Moreover, we emphasize that this argument relates to a **single parameter** $\alpha$ (as $\beta = 1-\alpha$). The capacity to overfit a dataset from tuning this one parameter is low.
> >
> > To control for this possibility, regardless, we performed hyperparameter sweeping of the scaling factors for Composable CoT + Merging and StandardCoT + Merging on zero-shot compositions of string manipulation and arithmetic tasks, following the same protocol described in Appendix G.4 but only using the atomic validation dataset for selection. We observed that the **same set of scaling factors** is selected using the atomic validation set and the composition validation set for each task-method pair.
> >
> > > The "quality" analysis checks whether both atomic CoT templates appear, not whether the model actually consumes the earlier step's outputs (step-level causal dependence or variable-passing).
> >
> > Thanks for the suggestion! We did observe in the traces that the model was using intermediate outputs, but hadn’t quantified it in the paper, so we provide an additional result here zooming in on this. We define intermediate accuracy as the percentage of test examples where the model responses: (1) show CoT patterns for both atomic CoTs; and (2) have the correct intermediate result. This metric measures how well the models can use both atomic CoTs and get step-level outputs for the earlier step. For the string manipulation and arithmetic tasks, the intermediate result is the answer to the first atomic question in the composition. For Skill-Mix, there is no variable-passing for such a writing task, so we consider the intermediate correctness as the percentage of skill-related phrases mentioned in the CoT that are also included in the final output sentence (this makes sure that the model does not just discuss two skills but excludes them in the final output).
> >
> > |                      |  Intermediate Accuracy |  |
> > | -------------------- | --------------------- | --------------------- |
> > |  | StandardCoT-MTL      | ComposableCoT-MTL     |
> > | Next Letter + Mult   | 0.0                   | 92.1 |
> > | Concat + Mult        | 22.1                  | 53.4 |
> > | Concat + Next Letter | 56.4                  | 72.7 |
> > | Skill-Mix            | 35.5                  | 38.4 |
> >
> > The table above summarizes the intermediate accuracy results of ComposableCoT and StandardCoT models. These results show that **Composable CoT models not only use both atomic CoT templates more frequently, but also produce correct outputs for the earlier step more frequently.**

---

> > > ### Author Response · Authors · 2025-11-21
> > >
> > > > Many findings are unsurprising and the novelty is modest. The main lift plausibly comes from making multi‑CoT sequences in‑distribution via formatting. The work’s practical impact beyond controlled compositions may be limited without evidence on untagged, naturally compositional tasks (e.g., program‑of‑thought planning, tool‑use pipelines).
> > >
> > > We argue that **controlled compositions evaluated in our paper are necessary for rigorous evaluation of compositional generalization with LLMs** for the following reasons.
> > >
> > > (1) **Atomic skills are distinguishable**: To evaluate compositional generalization, the atomic skills need to be distinguished from each other so that learning one skill is independent from learning another skill. Our task design aims at making the distinguishability as strong as possible.
> > >
> > > (2) **Compositions are unseen**: Given the size of the corpus LLMs are pre-trained on, it is hard to ensure that compositional examples drawn from general reasoning tasks, such as math word problems, are not similar to the data seen during pre-training. Our task designs aim at creating compositional examples that are less commonly seen in the pre-training data.
> > >
> > > Given the above considerations, we note that our task design has several advantages over other tasks, such as tool-use pipelines and program-of-thought planning. Moreover, our evaluation setting is not only limited to controlled compositions of string manipulation and arithmetic, but also includes a creative writing task (Skill-Mix) that captures aspects of real-world language understanding: models are required to write sentences using writing techniques, similar to what might happen in a co-writing model. We believe the suite of evaluation tasks used in this paper is representative enough for scientific understanding of compositional generalization of LLMs.
> > >
> > > [1] Deepseek-AI et al., 2025. DeepSeek-R1: Incentivizing Reasoning Capability in LLMs via Reinforcement Learning.
> > >
> > > [2] Muenninghoff et al., 2025. s1: Simple test-time scaling.
> > >
> > > [3] Schick et al., 2023. Toolformer: Language Models Can Teach Themselves to Use Tools.
> > >
> > > [4] Jin et al., 2025. Search-R1: Training LLMs to Reason and Leverage Search Engines with Reinforcement Learning.
> > >
> > > [5] Singh et al., 2025. Agentic Reasoning and Tool Integration for LLMs via Reinforcement Learning.

---

### Note · Authors · 2026-01-08

I have read and agree with the venue's withdrawal policy on behalf of myself and my co-authors.